# All You Can Feed: Some Comments on Production of Mouse Diets Used in Biomedical Research with Special Emphasis on Non-Alcoholic Fatty Liver Disease Research

**DOI:** 10.3390/nu12010163

**Published:** 2020-01-07

**Authors:** Sabine Weiskirchen, Katharina Weiper, René H. Tolba, Ralf Weiskirchen

**Affiliations:** 1Institute of Molecular Pathobiochemistry, Experimental Gene Therapy and Clinical Chemistry (IFMPEGKC), RWTH University Hospital Aachen, D-52074 Aachen, Germany; sweiskirchen@ukaachen.de (S.W.); k.weiper@web.de (K.W.); 2Institute of Laboratory Animal Science and Experimental Surgery, RWTH University Hospital Aachen, D-52074 Aachen, Germany; rtolba@ukaachen.de

**Keywords:** animal experimentation, diet, nutrition, ingredients, lard, fibers, fructose, diet coloring, autoclaving, irradiation

## Abstract

The laboratory mouse is the most common used mammalian research model in biomedical research. Usually these animals are maintained in germ-free, gnotobiotic, or specific-pathogen-free facilities. In these facilities, skilled staff takes care of the animals and scientists usually don’t pay much attention about the formulation and quality of diets the animals receive during normal breeding and keeping. However, mice have specific nutritional requirements that must be met to guarantee their potential to grow, reproduce and to respond to pathogens or diverse environmental stress situations evoked by handling and experimental interventions. Nowadays, mouse diets for research purposes are commercially manufactured in an industrial process, in which the safety of food products is addressed through the analysis and control of all biological and chemical materials used for the different diet formulations. Similar to human food, mouse diets must be prepared under good sanitary conditions and truthfully labeled to provide information of all ingredients. This is mandatory to guarantee reproducibility of animal studies. In this review, we summarize some information on mice research diets and general aspects of mouse nutrition including nutrient requirements of mice, leading manufacturers of diets, origin of nutrient compounds, and processing of feedstuffs for mice including dietary coloring, autoclaving and irradiation. Furthermore, we provide some critical views on the potential pitfalls that might result from faulty comparisons of grain-based diets with purified diets in the research data production resulting from confounding nutritional factors.

## 1. General Aspects of Mice in Biomedical Research

The laboratory mouse derived from the house mouse (*Mus musculus*) has been first used in biomedical research as a model system since the 17th century [1]. The earliest documentation of the use of mice in scientific research was done in the year 1664 in England, where Robert Hooke in his study used this animal model to study the biological consequences of an increase in air pressure [1]. In the 19th century, mice were used for a couple of breeding experiments, in which coat color or behavioral mutations were studied. Since that, these rodents have been used in many research areas. In 1981, a first genetically engineered transgenic mouse model was introduced that expressed the herpes simplex virus thymidine kinase [2].

Thereafter, both transgenic and knockout mouse models have become essential tools in the field of immunology, oncology, toxicology, genetics, and many more. These models allow the determination of the general consequences of alterations in individual genes and their cooperation with other genes. Particularly, inbred mice, which are “isogenic organisms” (nearly) identical to each other in genotype and phenotype, are frequently used for such studies. In respective experiments, these highly similar “linear test animals” are most suitable to establish reproducible results and conclusions. Moreover, testing in mice is a central part of drug development for humans, in which they are vital as a means for pre-clinical safety and efficacy testing before starting a human trial with a candidate drug. Therefore, it is not surprising that experimental work in mice has developed as an integral part of biomedical research in the building of basic knowledge. Exemplarily, mice experiments have developed as the gold standard to confirm a proposed disease-associated mechanism in hepatology research, in particular, non-alcoholic fatty liver disease (NAFLD) [3]. Specialized protocols have been developed closely mimicking typical clinical situations, including cholestasis, poisoning, metabolic injury, portal hypertension, inflammation, fibrosis, cirrhosis, and hepatocellular carcinoma [3]. During the last decade, the individual steps in such procedures were summarized in highly standard operating protocols (SOPs) to achieve uniformity in performance and outcome [4].

At the end of 2013, the seventh report on the statistics on the number of animals used for experimentation and other scientific purposes in the member states of the European Union (EU) was published [5]. This concise dispatch contains detailed information about the number of laboratory animals used for biomedical research in 2011 in the 27 member states with the exception of France. According to this report, a total of 11.5 million animals were used in biomedical research in 2011. From these, mice are the most commonly used species with 60.9% (~7 million animals) of the total use [5]. Although precise numbers for the worldwide total numbers of mice in biomedical research are not really available, first conservative estimates of annual laboratory animal use suggested at least 115.3 million animals to be sacrificed for scientific purposes in the year 2005 [6]. If globally the frequency of mouse usage in all kinds of animal studies is the same as in the EU (~61%), this means that about 70.2 million mice are annually sacrificed by scientists and clinical researchers. However, the strict implementation of the three Rs (replacement, reduction, and refinement) concept for animal experimentation proposed by Russell and Burch in 1959 [7] that is now mandatory standard in biomedical research in many countries [3], the introduction of relevant replacement methods [8], and finally the increasing political and public concern about animal experimentation influencing people‘s view toward the use of animals in research [9] have led to a significant reduction in animal number [5].

Nonetheless, all these laboratory mice bred for scientific purposes and kept in laboratory must be supplied with food. When estimating that each of the 70.2 million mice typically eat 3.5–3.75 g of food per day (10–15% of their body weight) [10] and assuming that the life span of a mouse in a typical setting of an Institute for Laboratory Animals Science is about six months (Tolba and Weiskirchen, unpublished), it can be calculated that about 44.23–47.39 million kg dietary food products are necessary to feed these mice. Surely, this value is a high underestimation because potential losses due to throwing away, passing of expiration dates, unwanted food spoilage, and many other circumstances have not been taken into account in this simplified calculation. Moreover, based on special requirements necessary in the individual mice experiments, a large variety of companies have developed dedicated to developing and providing products that meet the unique challenges for all kinds of experiments.

Such “special needs” are considered in products manufactured and marketed as “custom diets”. In comparison to “standards diets” or “chows”, these products are more expensive in formulation because they require costly dietary manipulation such as the addition of vitamins, minerals, special fats, cholesterol, proteins, dietary fibers, drugs or other compounds. Unfortunately, in the literature the terminology of diets is used somewhat inconsistently. Different reports use terms such as “mouse diet”, “rodent chow”, “custom diet”, “defined diet”, “purified chow”, “special diet”, “purified ingredient diet”, “grain-based diet”, “standard chow”, and many others. However, there are basically only two types of diets, namely the “grain-based diets” and “purified ingredient diets”. While “grain-based diets” are made out of grain, cereal ingredients, and animal by-products, “purified ingredients diets” are composed of highly refined ingredients [11].

In addition, rodent diets may require sterilization techniques when animals sensitive to normal or opportunistic microbes, such as immune compromised or germ free mice, are investigated. Food sterilization or decontamination is possible by exposure to γ-irradiation or by high-vacuum autoclaving. Highly sensitive diets can also be vacuum-, gas-, or modified atmosphere-packed (i.e., nitrogen-purged), which minimizes the risk of spoilage by oxidation.

Therefore, all these circumstances illustrate that the production of standard and customized diets intended to be used as mice feed is a complex business and a science in itself. We here will summarize some important issues on leading manufacturer, the production and diversity of mice research diets, their ingredients, and treatments during production.

## 2. Producers of Mouse Diets

Usually most laboratory mice are kept in centralized, well-designed, managed animal facilities, which allow efficient, economical and safe animal experiments. Depending on the design and size of the animal facility, the mice are either kept in high barrier, specified pathogen free (SPF) areas with restricted access to animal facility staff only or in low barrier (conventional) areas with additional access for licensed scientists. Usually, trained and skilled staff takes care of the animals and scientists usually don’t care about the formulation and quality of diets the animals receive during normal breeding and keeping. However, scientists investigating certain immunodeficient strains, analyzing diet-induced impairments, or conducting experiments in which nutritional factors interfere with the outcome of their experiments are more interested in the products feed to their mice.

Grain-based diets are commercially manufactured in an industrial process and the safety of products should be addressed through the analysis and control of all biological and chemical materials used in the production process. Companies with an international reputation for quality often are certified by quality assurance systems and work in strict accordance with the guidelines provided by either local (e.g., England: Food Standards Agency, FSA; France: French Agency for Food, Environmental and Occupational Health & Safety, ANSES) and/or international institutions such as the International Organization for Standardization (ISO), the European Commission (e.g., https://ec.europa.eu/food/safety/animal-feed_en) or the good manufacturing practices (GMP) of the World Health Organization (WHO). If necessary, these specifications must then be adapted locally by the responsible animal welfare authorities. These guidelines or directives ensure that manufacturing, testing processes, and labeling and batch processing record are clearly defined, validated, reviewed, and documented, providing the basis to conduct good laboratory practice (GLP) studies. Furthermore, these regulations guarantee that mouse diets are prepared under good sanitary conditions and truthfully labeled to provide information of all ingredients. As such, they are rather similar to the guideline used for human foods.

## 3. The Production Process

### 3.1. Grain-Based Diets

Grain-based diets are made in “closed formulas” containing natural ingredients such as soybean meal, ground corn, fish meal, animal byproducts, and very high levels of both soluble and insoluble fibers [12]. In addition, such chows frequently contain non-nutritive but biologically active compounds such as phytoestrogens and toxic heavy metals. Grain-based diets for biomedical research purposes are made in accredited facilities using SOPs that guide all facets of diet production. Each diet formula is manufactured by a fixed formula or are produced by supplementation of a “constant nutrition” designed and supervised by a nutritionist. Depending on the ingredients, fixed formulas can reduce variation of nutrients from batch-to batch. However, grain-based diets may still be subject to variation due to the complex nature of the ingredients in these diets, which contain multiple nutrients and non-nutrients known to be subject to variation. Different work steps are integrated into a linear sequence within the production process (Figure 1).

Key steps in the production process of grain-based diets in large quantities are the choice and delivery of raw materials (Figure 2A–C), quality control of these materials (Figure 2D–E), compilation and assignment of lot numbers (Figure 2F), and the computer-controlled mixing of individual compounds (Figure 2G–I) in suitable mixers. When producing an “extruded diet”, the mixed substances are then mixed with steam and hot water and fed into the extruder barrel and forced through the die opening to form a product in desired shape and size (Figure 2J–L). During this process the quality and moisture content is continually monitored and/or adjusted. The final product is then packed in multilayer paper bags or sacks (Figure 2M) and screened for unwanted stray metal particles by passing through an in-line metal detection capability (Figure 2N). The packed diets are then transported and stocked in suitable facilities with controlled temperature and humidity conditions to avoid food spoilage (Figure 2O). From there, the diets are quickly retrievable on demand.

### 3.2. Purified Diets

Purified diets are composed of refined ingredients that have undergone further processing and the composition is open to the researcher [12]. They usually contain a standardized and balanced quantity of proteins, carbohydrates, fats and fibers provided as a mixture of casein, corn starch, soybean oil and cellulose. Therefore, these diets should be always constant in composition from batch to batch. Importantly, they can be individually tailored with all kinds of compounds and further changed in regard to their relative amount of standard ingredients.

More specialized diets produced on request of the customer are usually prepared in much lower quantities. The production process is more laborious and the equipment used such as the mixers are much smaller (Figure 3A). Furthermore, most of these diets are irradiated and are delivered in smaller heat-sealed packages. To avoid mix-ups of these customized diets with other diets, these products are frequently colored with artificial food colors. Therefore, the appearance of these diets can be highly diverse (Figure 3B). Of course, such products are more expensive because of the labor-intensive work process and the often expensive ingredients requested by the customer.

High-fat diets (HFD) are widely used in studies of diet-induced obesity (DIO) and metabolic injury [13]. Diets enriched with different dietary fibers such as barley beta-glycan, apple pectin, inulin, inulin acetate ester, inulin propionate ester, inulin butyrate ester or combinations thereof have recently been shown to induce specific differences in cecal bacteria composition [14]. The beneficial effects of inulin-enriched diets and their modulatory role in microbiota composition were also sufficient to reduce inflammatory gene expression in hippocampus of APOE4 transgenic mice that develop systemic metabolic dysfunction and symptoms of Alzheimer’s disease [15]. In line, the supplementation with galactooligosaccharide improved the intestinal barrier in hyperlipidemic mice lacking the low-density lipoprotein receptor (LDLR) [16]. In mice, dietary fructose impairs mitochondrial size, function, and protein acetylation, thereby decreasing fatty acid oxidation and development of metabolic dysregulation [17]. Strikingly, the exposure of mice in utero and in the pre-weaning period to a methyl donor-supplemented diet provoking DNA methylation resulted in significant attenuation of repetitive motor behavior development that persisted through early adulthood [18].

All these examples show that the diet has tremendous impact on mouse health and disease and that profound changes in the composition of diets may affect the reproducibility or outcome of mice experimentations. The knowledge of the composition of a diet during an experiment is highly crucial to maximize experimental reproducibility requiring highly standardized conditions.

To reduce experimental variation among laboratories, the American Institute of Nutrition (AIN) established a committee in 1973 with the aim to establish general guidelines in preparing dietary standards for nutritional studies with laboratory rodents [19]. These recommendations should help scientists with limited experience in experimental nutrition and facilitate interpretation of results among experiments and laboratories [19,20]. The first diets introduced by the respective committee, i.e., AIN-76 and AIN-93, contained fixed formula supporting growth, reproduction and lactation [20]. Although subsequent changes in respective diets were introduced later, these nutritional guidelines are still applicable today and provide a global standard for purified mouse diets [19].

In line with these efforts, small size feed suppliers or larger companies accredited by the Association for Assessment and Accreditation of Laboratory Animal Care (AAALAC) or other councils produce and/or market standardized mouse diets with fixed formulation, which allow reducing the fluctuation in study results (Table 1).

## 4. Pasteurization

Pasteurization is a term named for the French scientist Louis Pasteur for a mild heat-treatment process used to destroy pathogenic microorganisms and preventing of spoilage of foods and beverages. For pasteurization of milk, for instance, a low-temperature, long-time process (LTLT) for 30 min heating at 63 °C or a 15 sec high-temperature short-time process (HTST) is used to inactivate large (but not all) spoilage-causing vegetative forms of microorganisms [21]. Beside LTLT and HTST, very short heating to 138 °C or above for at least 2 sec (ultra-pasteurization) is used for specialized applications [21].

Autoclaving referring to a process of sterilization under pressure is believed to be one of the most efficient methods of sterilization and common practice in many research institutions [22]. This method is inexpensive, convenient and guarantees the destruction of all microorganisms including spores and viruses. Therefore, pasteurization is commonly used for “sterilization” of mouse diets (Figure 4).

Usually, animal diets are autoclaved at 121 °C for 20 min, which can result in pellet hardness resulting in a significant reduction in wastage and in apparent and true consumption of the pelleted diet [23]. In addition, losses of pantothenate, vitamin A and vitamin D were found during autoclaving, while thiamine, riboflavin and pyridoxine were less affected [24]. In particular, autoclaving at higher temperatures for shorter period were more detrimental than autoclaving for longer time intervals at lower temperatures [24]. Therefore, feed fortification with vitamins is potentially necessary, especially after autoclaving at high temperatures to ensure that the maintained mice receive an adequate and balanced supply with these essential nutrients.

In addition, it is known that this procedure exerts undesirable effects on feed quality due to production of toxic compounds (e.g., acrylamide) and reduction of the overall nutritional value [22]. The autoclaving of a standard rodent diet resulted in a 14-fold increase in acrylamide, while the content of endogenous acrylamide in diets subjected to irradiation was reduced [25]. The forming of acrylamide is strongly correlated to the temperature used for sterilization [22,25]. In addition, autoclaved food products with high quantities of acrylamide produce elevated concentrations of epoxides, which are highly reactive chemicals, acting as mutagens [22]. Therefore, investigators and institutions should consider the detrimental and toxic effects that autoclaving might provoke in mouse diets.

## 5. Irradiation

In some cases, the diets are irradiated with γ-rays to eliminate remaining microorganisms residing in the feed. The microbial reduction strategies are often used to sterilize diets used for animals kept under SPF conditions [26]. In a typical sterilization process, the required dose depends on the “initial bioburden” and irradiation doses of between 20 and 30 kGrays (kGy) are used most frequently to treat diets intended for SPF animals, while larger doses (40–50 kGy) are recommended for diets intended for gnotobiotic or germ-free animals [26]. The physical unit Gy is defined as the absorption of one J of radiation energy per kg of irradiated material. For irradiation of large quantities of mouse diets, the pallets of product are loaded onto conveyors moving around the γ-ray source. In most of the commercial facilities for food irradiation, cobalt-60 (^60^Co) is the most common source of γ-rays. The respective facilities contain a number of safety systems, which are designed to avoid exposure of personnel to radiation. Furthermore, such irradiation devices are girded by a thick shield that hampers the penetration of γ-rays to the outside (Figure 5).

Although it is often argued that the doses used to destroy microorganisms are rather low, caution is advised because γ-rays at these doses have profound effects on some the integrity of the individual ingredients of the diet. This was impressively shown in a systematic study, in which the amounts of fat, protein, carbohydrate, and vitamins was investigated, showing that γ-rays at a dose of about 30 Gy have profound and selective effects on the stability of vitamin A and peroxide content of dry animal diets [26]. In line, a more previous study summarizing the main finding of published literature showed profound losses of the vitamins C, B_1_, E, and A in food after its irradiation [27]. Moreover, destruction of highly polyunsaturated fatty acids up to 98% and destruction of fatty acids with two double bonds up to 46% with accompanying lipid peroxide formation after irradiation with doses of 2–10 kGy [28], in which the effective dose of γ-rays refers to the amount of radiation that penetrate in the middle of the product to be irradiated.

## 6. General Remarks on the Origin of Nutrients in Purified Diets

In nature, mice have an extremely diverse diet, consuming practically any food source to which they have access. In order to be fed up, wild mice spend many of their active hours (~20 h) searching for these food products. Contrarily, the amount of time taken for a laboratory mouse to gnaw and eat well-balanced food directly from the cage hopper is considerably less [29]. Ideally, these preformed diet products are nutritionally either complete for various life stages from breeding through long-term maintenance, or adapted to special needs occurring during husbandry and housing. Global sold mouse diets contain a well-balanced mixture of proteins, carbohydrates, fats, vitamins, minerals, and potential additives that may be modified for the special needs of the biomedical research undertaken. In these formulations, substances that have been reported to have adverse confounding effects on experimental results or are toxic to the animals should as far as possible be omitted and should conform to the nutrient requirements of mice established by the National Research Council (see below). When properly stored at room temperature or cooler, depending on the composition of the diet, with ideally lower 50% relative humidity, these diets are usually stable for 6–12 months. To prevent continuous exposure to light and air, the storage in the original packaging or closed containers is recommended. Special diets enriched with temperature-sensitive additives may require the storage at lower temperature (4 °C or −20 °C). To avoid contamination, the products should be stocked in a proper environment.

The individual substance classes of which a diet is composed may originate from different sources (Table 2). The respective source may have an impact on energy intake, feed efficiency, apparent nitrogen and fat digestibility, composition of gut microbiota, and of course, of body weight development in mouse [30].

### 6.1. Proteins

Casein, soy protein isolate, egg white proteins often serve as sources for proteins in respective chows. Caseins are the most frequent protein constituent in animal milk from cow, sheep and buffalo containing with an intrinsically disordered structure forming large colloidal particles with calcium phosphate to form casein micelles [31]. It is enriched in proline, which distorts protein folding into α-helices and β-sheets preventing the formation of higher proportions of secondary and tertiary protein structures. However, casein proteins are important nutritionally because of their high phosphate content due to which they bind significant quantities of calcium ions [31].

Compared to casein, soy protein isolate has a hypocholesterolemic effect [32] due to a lower intestinal absorption of cholesterol, increased steroid excretion, and a greater biological activity in decreasing hepatic lipogenic enzymes [32]. As a consequence, mice fed soy protein isolate or soy protein isolate hydrolysate diets have lower body weight, lower plasma cholesterol and glucose levels compared to animals that are fed with a casein diet [32].

Egg white contains proteins high in amino acid balance, only low quantities of carbohydrate, and is almost free of fat [33]. Compared to other diets, the consumption of egg white in mice has no significant impact on total cholesterol, high density lipoprotein (HDL), low density lipoprotein (LDL), or triglyceride levels, and suppresses food intake, dietary fat absorption, and fat accumulation, thereby preventing the formation of glucose tolerance [34,35]. However, white egg supplementation is supposed to induce oral desensitization and immune tolerance in mice [36]. Nowadays, egg white as a protein source is not often used. Nevertheless, this protein source can be helpful in studies analyzing effects of zinc-deficient diets. This is due to the fact that the zinc-binding capacity of casein is about 8.4 µg–30 µg/mg casein [37,38], while the maximal amount of bound zinc in egg white products is estimated to be only 1.3–1.6 ng/mg [39]. Moreover, egg white contains large quantities of the anti-nutrient avidin having strong affinity for biotin, preventing its absorption across the gastrointestinal tract [40]. Therefore, supplementation with biotin is sometimes necessary when using egg white as the major protein source.

Chemically-defined diets containing crystalline amino acids as the sole source of nitrogen as an alternative to complete proteins have also been successfully used for mouse maintenance [41]. It has been shown already decades ago that amino acid rations, when properly compounded, will provide a rate of growth in mice closely approaching that obtained with casein [42].

### 6.2. Carbohydrates

It is well-known that mice become obese when offered free access to sugars, but it is not established whether specific sugars are more likely to cause DIO [43]. Most common in purified diets as sugar sources are sucrose, fructose, and corn starch. The disaccharide sucrose is a disaccharide composed of glucose and fructose produced naturally in plants and after oral uptake efficiently hydrolyzed by sucrose in the intestinal mucosa to its constituent monosaccharides [44]. Free glucose elicits a glycemic and insulinemic response that stimulate the uptake of this sugar into cells, while fructose is mainly metabolized in hepatocytes via insulin-independent mechanisms not regulated by energy supply [44]. Sucrose has an energy of 16.8 kJ (4 kcal) per gram. Interestingly, sucrose stimulates higher daily intakes than isocaloric fructose solution in mice [43]. In animals, the fruit monosaccharide fructose produces profound metabolic disturbances, including insulin resistance, impaired glucose tolerance, high insulin and triglyceride levels, hypertension, dyslipidemia, and microvascular hepatic steatosis [45]. However, the susceptibility to sugar-induced obesity varies with strain [45]. It has an energy density of 15.75 kJ/g (3.75 kcal/g) and feeding mice with a high fructose diet induces hepatic lipid accumulation by activating lipogenic gene expression and de novo lipogenesis [46]. Therefore, this sugar is supposed to be one of the key dietary catalysts in the development of non-alcoholic fatty liver disease [47]. Interestingly, elevated uptake of fructose in mice can result in dysbiosis, increased hepatic lymphocyte infiltration, and further inflammation of gut, liver and fat tissue [47].

Corn starch also known as “cornflour” is a glucose polymer, highly branched carbohydrate (e.g., the starch) derived from the endosperm of the kernel of corn (maize) grain. In its pure form it is a tasteless, odorless, and cold water insoluble powder. In the body, starch is hydrolyzed by amylases into its constituent sugars. Its energy content is about 15.95 kJ/g (3.8 kcal/g). Depending on its formulation, certain glucose polymers may resist digestion in the small intestine in mammals and arrive in the colon where they will be fermented by the gut microbiota resulting in a large variety of products including short chain fatty acids (acetate, propionate, butyrate) that provide as a prebiotics a range of physiological benefits [48]. This should be critically kept in mind when performing experimental studies analyzing the impact and composition of gut microbiota on energy homeostasis, development of obesity and its metabolic consequences [49]. Similarly, the feeding of C57BL/6J mice with HFD supplemented with resistant starch derived from maize resulted in an altered gut bacteria composition and corroborated with a significant shift in the liver metabolome [50].

### 6.3. Fats

A normal rodent diet contains about 10 kcal% fat, while diets enriched with 30–60 kcal% are defined as HFD, provoking significant weight gain and insulin resistance [51]. Typical fat constituents in mouse diets are lard, corn oil, safflower oil, or Menhaden oil.

#### 6.3.1. Lard

Lard is a semi-soft fat derived from adipose tissue of the pig and contains a high content in saturated fatty acid (~30%) and <1% trans-unsaturated fatty acids (i.e., trans fats). In some cases, lard is hydrogenated or treated with bleaching and deodorizing agents, emulsifiers, and antioxidants to improve its stability. The energy content of lard is about 37.6 kJ/g (9 kcal/g). Interestingly, in rodents lard-based HFD accentuated the increase in weight gain and the development of obesity and insulin resistance more than a diet that was based on hydrogenated vegetable-shortening diets, suggesting that the outcome of consuming HFD is strongly dependent on the used fat constituent [52]. Moreover, lard-based diets were significantly more inferior than soybean oil in protecting mice after application of the powerful hepatotoxin carbon tetrachloride twice a week for three weeks, which is a model to generate liver necrosis and steatosis, potentially indicating its less antioxidant activity [53].

#### 6.3.2. Corn Oil

Refined corn oil is derived from the germ of maize and typically contains 99% triacylglycerols with 59% polyunsaturated fatty acid (e.g., linoleic acid), 24% monounsaturated fatty acid (e.g., oleic acid), and 13% saturated fatty acid (e.g., palmitic acid, stearic acid, arachidic acid) [54,55]. It is categorized as one of the richest sources of health-promoting phytosterols and tocopherols protecting against DNA damage, hypertension, platelet aggregation, hypercholesterinemia, and diabetes [54]. In line with these beneficial effects, high corn oil dietary intake was shown to improve health and longevity of aging mice when fed at normal energy balance [56]. In addition, disease development and progression as well as deposition of extracellular matrix within the liver in a mouse model of non-alcoholic steatohepatitis (NASH) was significantly reduced when the HFD was composed of corn oil instead of non-trans fats [57].

As mentioned, the dietary intake of corn oil is known to improve health and longevity of mice, which corroborates with reversing aging-increased blood lipids and decreasing serum pro-inflammatory markers [56]. In addition, the olfactory cues and the oily texture of corn oil are important orosensory factors provoking a strong appetite in mice [57]. However, other reports showed that the excess dietary intake of polyunsaturated fatty acids is associated with loss of spontaneous physical activity and development of insulin resistance [58]. In addition, polyunsaturated fatty acids are subject to oxidation. Therefore, the AIN recommended the supplementation of antioxidants in formulations containing large quantities of corn oil [19].

#### 6.3.3. Safflower Oil

The safflower (*Carthamus tinctorius*) or safflor is a thistle-like annual plant of the *Asteracea* family from which vegetable oil can be extracted from its seeds. Safflower seed oil is flavorless and colorless and in its composition similar to oil from sunflowers, olives, and peanuts, typically containing high content of linoleic acid (63–72%), oleic acid (16–25%) and linolenic acid (1–6%) [59]. In particular, the high content of linoleic acid was shown to have highly beneficial health-promoting effects by reducing the expression of lipogenic enzymes and increasing the activity of hepatic fatty acid oxidation enzymes [60].

#### 6.3.4. Menhaden Oil

The forage fish menhaden (*Brevoortia tyrannus*) belongs to the herring family and forms large flocks occurring on the North American Atlantic coast from Nova Scotia to Florida and related forms are also found up to the coasts of Argentina. The oil derived of these animals is rich in omega-3 polyunsaturated fatty acids such as eicosapentaenoic acid (EPAc) and docosyhexaenoic acid (DHAc), both supposed to have anti-inflammatory activities [61]. Recently, it was demonstrated that EPAc and DHAc supplementation in the context of HFD partially mitigated reductions in insulin sensitivity and maintaining cell function [62]. Moreover, the polyunsaturated fatty acids in menhaden oil prevented high-fat diet-induced fatty liver disease in mice [63].

### 6.4. Vitamins

Mice, like humans, require some essential micronutrients in small quantities that cannot be synthesized by their own. These vitamins are organic molecules and must be obtained through the diet, or alternatively synthesized by microorganisms in the gut flora. They have diverse biochemical functions and are commonly sub-classified as either water-soluble (vitamin C, vitamin B_1_, vitamin B_2_, vitamin B_3_, vitamin B_5_, vitamin B_6_, vitamin B_7_, vitamin B_9_, vitamin B_12_) or fat-soluble factors (vitamin A, vitamin D, vitamin E, vitamin K). In comparison to fat-soluble vitamins that can accumulate in the body, water-soluble vitamins are readily excreted from the body. Deficient intake (primary deficiency), malfunction during absorption or use of a vitamin (secondary deficiency), or increased consumption results in hypovitaminosis. In contrast, excess intake results in hypervitaminosis occurring mainly only with fat-soluble vitamins (e.g., vitamin A and D). The different vitamins are involved in many biochemical processes (Table 3). Therefore, any shortage might result in complex illnesses potentially affecting different organs. General guidelines defining the nutrient requirements of the mouse are available (see below) [10].

Based on the heterogeneous character of the different vitamins, their stability is highly variable. Quantitatively deterioration in content over time of vitamins can be affected by many factors, including temperature, moisture, oxygen, light, pH, oxidizing and reducing agents, catalytic activity of metals, mutual damage by other vitamins, detrimental compounds (e.g., sulphur dioxide), or combination of these factors [79]. For example, vitamin B_12_ is decomposed by light, alkali, acids, and oxidizing or reducing agents, while on the contrary vitamin B_2_ (riboflavin) and vitamin B_3_ (niacin) are rather stable [79].

In addition, during production and handling of mouse diets, there are several factors affecting the stability of vitamins during extrusion. These occur for example during handling of raw material, mixing, conditioning, processes, changes in moisture, heat or pressure treatments during extrusion and expansion [80]. Aspects of stability of vitamins and reduced levels of vitamins during processing of fish feed were concisely discussed by Riaz and coworkers [80]. Since the production of grain-based diets is rather similar, the reported values should be comparable with these values. For the different vitamins, the factors affecting vitamin destruction during processing and storage are different (Table 4). Sufficient supply with vitamins in mouse diets can be guaranteed by food fortification.

Information of this table was taken in simplified and modified form from [80] and complemented with data from [79,81].

### 6.5. Minerals and Trace Elements

Minerals, also known as macrominerals or micronutrients, are inorganic elements which generally occur in large quantities, while trace elements or microminerals normally are present only in small amounts in organisms. Calcium, phosphorus, chloride, magnesium, phosphorus, potassium, sodium, and chloride are the elements playing vital roles in the body [82]. They regulate the proper composition and function of the body fluids, tissue, bone, teeth, muscles and nerves. Some of them also have function as a coenzyme in metabolic reactions and guarantee biochemical functions in body homeostasis, including energy production, growth, wound healing and proper utilization of vitamins and other nutrients. Essential trace elements required in smaller amounts by animals and plants are iron, zinc, copper, nickel, molybdenum, manganese, selenium, iodine, and others. These elements are involved in vital enzymatic reactions by acting as cofactors or by stabilizing cellular structures [82]. Based on their involvement in hundreds of biological processes, inadequate mineral and trace element intake can result in severe health conditions that can affect nearly all organs and tissues. These can be highly variable and become most evident after chronic shortage (see below).

### 6.6. Fibers

In its simplest definition, fibers are non-starch polysaccharides composed of a large number of monosaccharides that are linked through covalent bonds [83]. The term “dietary fibers” is often used to designate the sum of non-starch polysaccharides with its complex fibrous, tasteless organic polymers (i.e., the lignin) forming key structural materials in the supportive tissue of plants of which it is derived of [83]. Based on its composition, these dietary fibers have different physiochemical properties regarding size, hydration, viscosity, fermentability, and impact on satiety. In addition, the proportion of the cell wall components varies from plant to plant and is further dependent on the age and type of plant tissue.

Dietary fibers are roughly grouped into soluble and insoluble fibers. Soluble fibers (non-cellulosic polysaccharides, arabinoxylans, β-glucans, some hemicelluloses, pectins, gums, mucilages, inulin) dissolves in water and are broken down in the gut some of which form a thick, spread-out gel, while insoluble fibers (cellulose, some hemicelluloses, lignin, resistant starch) are left intact as food moves through the gastrointestinal tract [84]. Some soluble fibers block the uptake of fats and are used as a fermentable energy source for gut bacteria. On the contrary, insoluble fibers are indigestible and speed up the elimination of toxic waste in the digestive tract through promoting bowel movement in the colon, thereby preventing constipation.

Fibers can be further sub-classified as neutral detergent fiber (NDF) and acid detergent fiber (ADF), in which NDF is the complete fraction of insoluble residue following neutral detergent digestion and ADF is the harder to digest part of the fiber. In other words, ADF is the sum of cellulose and lignin and NDF is the sum of ADF and hemicellulose. ADF is the fraction of fibers that contain virtually no fermentable ability and reduces overall digestible energy from the diet and NDF is a measure of most of the fiber in the diet (except for soluble fiber, which is not part of this fraction). Therefore, a high content of ADF in a diet will provide lower amounts of energy than a diet with lower ADF amount [85].

There are a large number of fiber sources used to dilute the nutrient and energy density of the diets (cf. Table 2). When fibers are included in rodent chows, the weight of the cecum and colon may increase and microbial fermentation results in short-chain fatty acid (SCFA) production such as acetate, propionate and butyrate having beneficial effects on mice health [86,87]. Moreover, addition of fibers that dilute the nutrient density of the diet will have effects on food intake and body weight and further impact the fecal and urinary nitrogen excretion as a result of microbial fermentation [10]. In humans, diets with a high content of fibers are suggested to have beneficial effects, including increasing the volume of fecal bulk, decreasing the overall time used for intestinal transit, promoting the elimination of toxic waste, stimulating the intestinal flora, and finally reducing the onset risk of metabolic syndromes (Figure 6) [88]. Comparable to humans, a low-fiber diet was shown to promote expansion and activity of colonic mucus-degrading bacteria, suggesting respective diets are ideally suited as nutritional models for analyzing aspects of colonic mucus layer dysfunction and altered pathogen susceptibility [89].

## 7. Nutrient Requirements of the Mouse

Mice as humans need a balanced, fresh and healthy diet that meets their nutritional needs. Nutrients designed for rats, guinea pigs, hamsters or other herbivores are not necessarily suitable for mice, because they need sufficient quantities of essential amino acids, fatty acids, vitamins, and minerals that might vary in content to other animals.

When housed under a standard 12-h light/12-h dark cycle, mice typically consume the majority of their food during the dark period, with short bouts of feeding during the light period [90]. There are a number of factors impacting food uptake, including strain differences, genetic background in transgenic and knockout mice, age, stress, habituation, forced movement, and discomfort resulting from drug treatments or surgeries. Moreover, the energy balance in female mice is strongly affected by hormonal variation associated with the estrous cycle [90].

Detailed guidelines for the nutrient requirements of laboratory animals were first published in 1962 and updated several times [10]. In the most recent edition, published in 1995, the composition of an adequate nutrition of mice maintained in conventional animal facilities is in detailed listed (Table 5). It should be noted that mice kept in a germ-free SPF facility or subjected to experimental-induced stress have altered nutrient requirements that should be adapted accordingly.

It was demonstrated some years ago that the composition of the commensal gut microbiota in humans correlates with diet and health in the elderly [91]. Moreover, in aged mice some of the alterations associated with aging can be rescued by fecal transfer [92]. In this context, it should be noted that the eating of fresh feces, which is a natural behavior of mice, is possibly not only helpful to better absorb nutrients/minerals they need to stay healthy, but is further a requirement to slow down the aging processes caused by nutritional deficiencies. The consequences of a chronic shortage in a specific nutritional compound have been best documented in mouse studies in which diets were fed lacking individual substances. Such studies have shown that the permanent lack in a specific component evokes severe consequences (Table 6).

The resulting phenotypes resulting from chronic shortage in specific elements or compounds can vary dramatically. The nutritional status impacts growth, reproduction, longevity and determines the response to pathogens, environmental stress, and organ function. Therefore, the avoidance of inadequate intake is one important factor in guaranteeing the welfare of the mouse kept in an animal facility.

## 8. Representative Examples of Diet-Induced Obesity and Fatty Liver Disease

The composition of a diet has strong impact on the health of an organism. It influences the composition of the gut microbiota and overfeeding or fasting can cause disease. Therefore, scientists frequently use such model to analyze aspects of diet-related diseases. On the contrary, lifelong caloric restriction is an effective experimental tool to reprogram hepatic fat metabolism and to extend life span in diverse species [126].

In biomedical research, mice are the most widely used animals and the nutrient requirements might depend on development state, reproductive activity, age, and stress factors induced by the experimental conditions. Moreover, there is a great danger that mice, when housed in standard laboratory under ad libitum feeding conditions having continuous access to food but virtually no environmental stimulation become overfed and sedentary and are potentially not suitable as proper controls in animal experiments [127]. Similarly, unwanted contaminants such as pesticides, mycotoxins, heavy metals, nitrosamines, nitrates, nitrites, phytoestrogens with estrogenic activity, and polychlorinated biphenyls in dietary products may affect the outcome of animal studies when present at a sufficient high concentration [128]. Maximum allowable concentrations of these undesirable substances in mouse diets are also specified by the guidelines provided by organizations such as the US Environmental Protection Agency (EPA), the Food and Drug Administration (FDA), the British Association for Research Quality Assurance (BARQA), or the Society of Laboratory Animal Science, GV-SOLAS [128].

Therefore, depending on the research question, the nutritional requirements must be carefully considered. Contrarily, the usage of specific diets in mice is widely applied to induce diseases mimicking human pathologies including liver disease, metabolic dysfunctions (insulin resistance, diabetes type 2), heart failures, immune system alterations, neurological disorders, or even cancer. These diet-induced models are enriched in specific fats, sugars, toxins, metals, or alternatively lack essential nutrients that are indispensable for the proper synthesis of essential nutrients. In particular, studies analyzing aspects of the immune system require special needs in regard to sterility and compounds that might interfere with the composition of the gut microbiome or function.

In most countries, the feeding of diets provoking the formation of disease or animal suffering require permission of responsible animal welfare authorities and should be carried out in an ethical framework that minimize fear, pain, stress and suffering of animals. This is best done, when respective experiments are carried out following established SOPs providing details about the scientific background, its implementation, experimental details (handling, concentrations, duration of procedure, biometric aspects, readout systems), and about the animal burden associated with this procedure [4]. The diversity of mouse diets is extremely versatile. The diets might vary in size, form, color and of course in nutritional composition (Figure 7).

Nowadays, a large number of diet-induced disease models are established in biomedical research. Most common are models to induce atherosclerosis, obesity, diabetes type 2, or liver damage. In the following, we will discuss some aspects of representative diets used in hepatology research, in particular, NAFLD. These are a typical control diet and four diets that are used to induce fatty liver injury, namely a diet to induce DIO, a typical Western diet (WD), a diet rich in fat, fructose and cholesterol (FFC), and the so called methionine-choline deficient (MCD) diet. Representative compositions of such diets are given in Table 7.

These four diets are some examples of diets commonly used in experimental hepatology research as nutritional models to induce a spectrum of disorders associated with accumulation of excess fat in the liver. The most common form is NAFLD and a more serious condition named non-alcoholic steatohepatitis (NASH). NAFLD and NASH and are the most prevalent liver diseases in Western society and the third leading cause for liver transplantation in the US [129]. Furthermore, there is evidence that NAFLD precedes and is associated with the metabolic syndrome characterized by obesity, diabetes, insulin resistance, and hypertension [130]. Phenotypically, patients with NASH/NAFLD are characterized by liver cell injury and damage, inflammation, and an increased risk for liver fibrosis and carcinogenesis [129]. Based on the eminent importance of NAFLD, several experimental dietary mouse models were developed to mimic the pathogenesis of human NASH and NAFLD.

When comparing a diet used for DIO with a control diet, the most striking difference is the high-fat content of the DIO diet (cf. Table 7). Typically, mice fed a DIO diet containing 40–60% of calories from fat for 7–30 weeks increases their body weight and propensity to develop pre-diabetic symptoms and metabolic syndrome. This type of diet is, therefore, often used in studies investigating aspects of food intake, energy expenditure, glucose tolerance, insulin resistance, and elevated blood pressure [130,131]. When using this model, a slight increase in body weight can be noticed already after 2–4 weeks, while the body weight gradually increases thereafter and is 20–30% higher in mice after 16–20 weeks compared to chow-fed mouse [132]. However, the outcome of the DIO model is influenced by many factors, including genetic background, gender, age, and environmental factors such as cage placement, mice density, and mice handling [132].

While in a typical control diet, the fat and sugar content is not higher than 10%, a WD is characterized by a high content of fat combined with a high amount of a sugar as sucrose or fructose. In some cases, these diets are enriched with trace of SCFA such as C4:0 (butyric acid) and medium-chain fatty acids (MCFA) such as C6:0 (caproic acid), C8:0 (caprylic acid), and C10:0 (capric acid) are added to these diets. The rationale of this supplementation is the notion that ghrelin activation requires acetylation of its third residue, serine, with caprylic acid by ghrelin O-acyltransferase [133].

After prolonged feeding of a DIO diet or WD to mice for 30–50 weeks, the animals become severely obese, fat deposition occurs, ectopic fat accumulates in the body and the liver size significantly increases (Figure 8).

In cardiovascular research, WDs enriched in cholesterol, cholate, sucrose, and/or saturated fatty acids have also atherogenic effects and are frequently used to induce or accelerate atherosclerosis in mice [134,135].

Diets highly enriched in fructose rapidly induce an early diabetic state in mice [136,137]. In the liver fructose can be converted in several steps to glycerol-3-phosphate and metabolized by de novo lipogenesis to fatty acids, which can then be esterified to triglycerides (Figure 9A). Therefore, the chronic intake of excess dietary fructose leads to increased formation of triglycerides that accumulate in the liver (Figure 9B), insulin resistance and formation of very low density lipoprotein, attributes that are hallmarks in NAFLD [46].

Interestingly, fructose-induced steatosis and damage induced by feeding a diet enriched in 60% fructose for four to eight weeks was more severe in female than in male mice, suggesting that respective diets provoke gender-specific differences during progression of disease [47]. Feeding of fructose in combination with fat and cholesterol for four days was already sufficient to induce hepatic triglyceride accumulation demonstrating that individual “unhealthy” compounds within a diet can be additive or synergistic [138]. Moreover, the feeding of fructose (60%) for four to eight weeks provoked impairment of olfactory epithelium, resulting in reduced olfactory behavioral capacities [139].

Other diets are characterized by the lack of essential components. In the MCD, sulfur-containing supplements are missing that cannot be synthesized de novo. When missing the essential amino acid methionine, S-adenosylmethionine (SAM or AdoMet) representing a common co-substrate involved in transmethylation, transsufuration, aminopropylation that further blunts inflammatory reactions, cannot be synthesized [140]. The lack of this compound results in lower quantities of cysteine, lecithin, phosphatidylcholine and many other macromolecules (Figure 10A) provoking significant fat accumulation and fibrosis progression in liver (Figure 10B).

Chronic shortage in methionine is, therefore, associated with a progressive physiopathology characterized by increased oxidative stress, hepatic upregulation of pro-inflammatory and pro-fibrogenic genes, liver damage as indicated by increased levels of aminotransferases, and manifestation of other NASH-associated symptoms [13]. Similarly, a shortage in choline, which is an integral part of phosphatidylcholine, sphingomyelin, and acetylcholine, results in significant intrahepatic lipid accumulation through a decreased production of very low density lipoproteins (VLDL), down-regulation of key enzymes involved in triglyceride synthesis, and impaired *de novo* lipogenesis [13]. As a consequence, harmful reactive oxygen species (ROS) are generated and the inefficient β-oxidation causes ballooning of hepatocytes, diffuse necrosis, and hepatic fibrogenesis, and on long-term liver cancer [13,141].

Cholesterol-enriched diets are widely used in studies investigating aspects of the metabolic syndrome. When mice were fed with a high (1%) cholesterol diet for 12 weeks, animals developed hyperlipidemia, hyperinsulinemia, and showed hepatocyte hypertrophy with extensive intracellular accumulation of lipid vacuoles and droplets [142]. It is suggested that in atherogenic diets, which are enriched for example in cholesterol and cholic acid, cholesterol is the key component driving oxidative stress resulting in steatohepatitis and insulin resistance [143]. In addition, these diets induced immune-related responses that may be related to liver damage in 12 inbred mouse strains tested [144].

In sum, these examples demonstrate that “unhealthy” diets enriched in or lacking of ingredients usually part of a balanced diet are suitable to provoke hepatic damage. Therefore, these diets are most popular in biomedical research to investigate mechanisms of initiation and progression of liver disease. However, many of these studies draw conclusions by comparing health aspects of animals fed a grain-based diet with a purified diet such as HFD. However, the effects of the dietary fat will be confounded with the effects of other components that differ between the diets. This fact has been already critically highlighted twelve years ago in a thought-provoking commentary in which 35 studies published in five prestigious high-impact journals were critically evaluated in regard to their performance [145] and this trend has continued as demonstrated by a more recent survey of a larger sampling of the same journals [11]. This exemplarily illustrates the fact that it is critical to draw conclusions when comparing dietary effects obtained in animals receiving either “grain-based diets” or “purified diets”. Although diets are normally produced in fixed formulation, minor differences might also result when comparing findings obtained with diets produced by different companies. However, these variations should be relatively negligible.

## 9. Special Ingredients

For some studies, mouse diets are fortified with special ingredients (Figure 11). Since the mid-1990s many genetically modified mice were developed, in which the transgene is directed under the control of a tetracycline (Tet)-dependent regulatory system [146]. In these “Tet-on” or “Tet-off” systems, doxycycline is preferable as an inducer in these systems due to its high biological potency, excellent tissue penetration, and its widespread availability [146]. This compound is rather stable in food products and its concentration is not significantly influenced by storage at room temperature or by exposure to light [146].

In other genetically-modified mouse systems, proteins are expressed as fusions with a modified estrogen receptor ligand binding domain. In these systems, the binding of this moiety to tamoxifen results in a conformational change that allows the fusion protein to translocate to the nucleus. The nuclear translocation of a dominant active transcription factor induces transcriptional activation of susceptible genes, while the expression of a dominant negative receptor fusion might provoke silencing of respective genes. Using this concept, also several inducible tamoxifen-dependent Cre recombinases (from English causes recombination) such as the estrogen receptor-dependent recombinase (CreER recombinases) were cloned that are widely used in biomedical research. They can direct the excision of LoxP-flanked DNA to generate genome modifications in mice [147]. However, using these models and respective diets enriched in tamoxifen, it should be noticed that this drug could influence locomotor activity, social interaction and anxiety in mice requiring critical planning of experimental design [148].

The isoflavone genistein is a phytoestrogen with antioxidant activity targeting numerous intracellular targets leading to retardation of atherogenic activity, possessing suppressive effects on both the cell-mediated and humoral components of the adaptive immune system, and inhibiting cancer progression by inducing apoptosis or inhibiting proliferation [149]. On the molecular level, it was shown that this compound inhibits a large number of enzymes, including adenosine triphosphate (ATP)-utilizing enzymes such as tyrosine-specific protein kinases, topoisomerase II and enzymes involved in phosphatidylinositol turnover [150]. Moreover, this substance has anti-angiogenic effects, modulates estrogen activity, and impacts DNA methylation and/or chromatin modification [151]. Based on this complex repertoire of activities, many mice studies have been performed with genistein-enriched diets. In one study, in which aspects of energy expenditure were analyzed in obese mice, 600 mg genistein/kg diet fed for a period of four weeks resulted in significantly increased food consumption without affecting body weight [152]. In a WD, the supplementation of 1.5 g genistein/kg diet decreased mouse food intake, body weight, and improved glucose metabolism [153]. In a study analyzing the impact of genistein on DNA methylation a concentration of 300 mg genistein/kg diet for four weeks was applied [154]. In the respective investigation, it was shown that male mice fed a casein-based diet containing genistein showed significantly higher DNA methylation in prostate than control male mice [154].

Similar to the biological effects of genistein on energy expenditure, diets enriched with a related isoflavone daidzein-rich isoflavone aglycone extract at 0.6% of the diet was able to reduce HFD induced body weight gain via reduced hepatic production of triglycerides and subsequent reduction of adipose tissue mass [155]. It was, therefore, suggested that the beneficial effects of daidzein and genistein on food intake and body weight gain in mice are mediated by alterations within the liver X receptor (LXR) signaling pathway [153].

The sugar myo-inositol representing one of nine distinct stereoisomers of inositol is a structural component of many second messengers and lipids such as phosphatidylinositol and its derivatives. Interestingly, when chronically given, this substance has insulin-sensitizing potential in mice provoking a significant decrease in white adipose tissue [156]. Diets enriched in myo-inositol at 2.64 g myo-inositol/kg diet also showed potent reduction in the number, size, and stage of lesions in cancer-prone transgenic mice through triggering alterations in macrophage recruitment and phenotype switching [157].

These examples show that special diets have become an essential research tool in biomedical research. The companies specialized in the production of rodent diets offer limitless custom formulated diets for virtually each application. The fields of applications are numerous, including addition of special fibers, fat, or sugars to modulate the murine microbiome, incorporation of metals or environmental toxins such as microplastic to test their toxicity, feeding of “drug diets” to test the safety of compounds, and many others. The mentioned representative examples of special ingredients demonstrate the high diversity that is possible to modulate diets for a specific purpose. Custom research diets spiked with substances or lacking essential compounds are one critical puzzle piece of biomedical research that help to unravel individual risk factors contributing to disease formation. Usually these diets are produced in small quantities are formulated after consultation with the manufacturers.

## 10. Diet Coloring

Food colorants can be divided into three groups: (i) Naturally-derived colors used for food coloring may originate from crushed insects (e.g., carmine), saffron, turmeric, carrot, beet or their color-making ingredients, such as riboflavin and β-carotene. These can be extracted with or without intermediate of final change of identity from biological sources. The addition of these colors to foodstuffs only needs to be approved in the country in which the product is sold or manufactured. (ii) In addition, several mineral or synthetic inorganic colors such as iron oxide, titanium dioxide, chromium oxides are certified as natural food colorings, or for use in drugs, cosmetics, medical devices, or animal food. In particular iron oxides black, red and yellow are intended to be used as colorings and restore color to animal feeding stuffs at a recommended concentration between 500 and 1200 mg/kg without posing a risk to the environment [158]. Since these dyes are highly stable and excreted essentially unchanged in the faces of the animal, they are also considered as safe additives. In many cases, they can be added without requiring to be certified by regulatory bodies when applied in amounts not exceeding preset maximum concentrations. (iii) On the contrary, artificial food colors also categorized as synthetic food dyes or certified color additives are dyes produced by chemical synthesis. In the US, these synthetic compounds must be approved for their usage by the FDA. Once approved, these food dyes are typically named by Federal Food, Drug and Cosmetic Act (FD&C) numbers, while in the European Union and Switzerland certified color additives are classified with European (E) numbers. In some cases, these synthetic dyes are classified as “coal-tar colors” because they were originally produced from petroleum or coal. Actually, the usage of seven colorings and their lakes are permitted in food products in the US (Table 8, Figure 12).

Usually the dyes used in rodent diets are applied as ionic salts rendered partially insoluble by interaction with a metal such as calcium or aluminum. The FDA defines these water-soluble dyes as “lakes”, in which the proportions of dye to metal are more or less fixed and given in percentages (e.g., FD&C Red No. 40, aluminum lake, 36–42%).

Commonly, the dyes for coloring of food are used to modify the appeal of food for humans [159]. However, in comparison to humans, mice do not have the visual ability to distinguish the abundance of colors [160]. Instead, the uptake of food by a mouse is strongly dependent on smell perception and olfactory system that are extremely important for controlling energy homeostasis [161]. The coloring of mouse diets is, therefore, in principle of no importance for the animals. However, the coloring has some useful properties. They allow the researcher to distinguish one diet from another and they ensure that contamination and transmissions to other diet products during the production pipelines can be recognized. In the following, we will give some short information about the artificial colors used for food coloring.

Brilliant blue (FD&C Blue No. 1, E133) is a reddish-blue triarylmethane water-soluble dye used as a blue colorant. It has reasonable stability when exposed to light, heat and acidic conditions, but it has overall low oxidative stability [162]. This dye is considered harmless and most of the dye is excreted undigested [163]. There is no evidence that this dye in rats or mice is carcinogenic, or neurotoxic [164]. However, this dye can act as purinergic inhibitor without pharmacological selectivity, thereby modulating some organ and tissue functions [162]. In addition, a recent report has demonstrated that this dye showed significant greater absorption in septic patients with reduced intestinal barrier function [165]. The daily maximum FDA-approved uptake of Brilliant blue for humans is 12.5 mg/kg body weight/day, while the EU scientific committee suggested an acceptable daily intake (ADI) of 10 mg/kg body weight/day [162].

Indigotine (FD&C Blue No. 2, E132) or indigo is a dark blue water-insoluble anionic pyrrole-based dye originally isolated from the leaves of certain tropical plants. Nowadays this dye is one of the most used coloring agents in the textile industry and synthesized by various methods [166]. In toxicity studies, indigo carmine, representing an organic water-soluble salt derived from indigo by sulflonation, showed no genotoxicity, developmental toxicity or modification of hematological parameters. An ADI up to 5 mg/kg body weight is presently considered as harmless [167]. Moreover, in traditional Chinese medicine, indigo as “Qing-Dai” alone or in combination with other compounds is used as an overall safe and effective drug for treatment of sun stroke, convulsions associated with epilepsy, cough, chest pain, hemoptysis, and phlegm and childrens convulsions [167]. In rodents, the majority of this dye is not absorbed, but readily broken down in the gastrointestinal tract to 5-sulfoanthranilic acid that is absorbed and excreted mostly in the urine [164].

Fast Green FCF (FD&C Green No. 3, E143) is a FDA-approved triphenylmethane dye, while its usage as a food dye is prohibited in the EU [168]. When administered orally 200 mg of this dye to rats, the dye was excreted unchanged in the faeces and no dye was found in the urine [169]. Mice fed diets containing up to 2% Fast Green FCF for 78 weeks, showed no lesions attributed to feeding of the color [170]. The estimate of temporary ADI for man is set up to 12.5 mg/kg body weight [171].

Erythrosine (FD&C Red No. 3, E127) is a cherry-pink poly-iodinated xanthene used as artificial red colorant in foods, drugs and cosmetics [172]. In the past, this dye was commonly used in many countries but is less commonly used in the US, where it is most often replaced by Allura Red AC (FD&C Red No. 40). Chronic toxicity and carcinogenicity studies performed in rats and mice revealed an increased incidence of thyroid follicular cell hyperplasia and adenomas in animals that received 4% erythrosine in the diet for 30 months following in utnero exposure [173]. However, this dye is non-mutagenic and thus the observed tumorigenic activity is most likely not the result of genotoxic initiation [174]. However, the dye has negative effects on thyroid function and therefore the temporary ADI is only in the range of 0–0.05 mg/kg body weight [175].

Allura Red AC (FD&C Red No. 40, E129) is a highly popular red azo dye that may cause allergic reaction such as urticaria or asthma, especially when administered together with orther synthetic color additives [167]. However, in general this dye at 0–7 mg/kg of body weight per day is considered as safe [167]. In the US population, Allura Red AC belongs to the three highest cumulative eaters-only exposures of FD&C color additives in food products [176,177]. Although the European food safety authority expressed concerns about the usage of Allura Red AC as a food color additive, the dye has no genotoxic activity in different test systems [178].

Tartrazine (FD&C Yellow No. 5, E102) is a water-soluble yellow monoazo dye used all over the world for food coloring. In a community-based, double-blinded, placebo-controlled food study, this dye in a mix with other artificial color additives provoked increased hyperactivity in young children [179]. However, the compound has an overall low toxicity with and LD_50_ value of greater than 2 g/kg body weight and an ADI for humans of 0–7.5 mg/kg body weight was established [177]. Similarly, in mice high-dose level in excess of this ADI were shown to produce only a few adverse effects in neurobehavioural parameters during the lactation period that were however unlikely produce any adverse effects in humans [180].

Sunset Yellow FCF (FD&C Yellow No. 6, E110) is an orange azo dye supposed to have no carcinogenicity, genotoxicity, or developmental toxicity in mice [181]. According to the WHO/FAO guidelines, the ADI was increased in year 2014 from 0–1 mg/kg body weight per day to 0–4 mg/kg body weight [182]. When high content of Sunset Yellow FCF (up to 5% for 23 months) were fed to mice, the mortality rate was not significantly different than in mice receiving no dye and the histopathological changes in organ and tissue observed were considered unrelated to the dietary administration of Sunset Yellow FCF [182].

Citrus Red (Citrus Red 2, E121) is an yellow to orange dye. Testings in mice showed that the feeding of diets containing 3% Citurs Red 2 caused increased morbidity and mortality in both sexes [183]. Based on a number of similar reports suggesting that Citrus Red 2 has carcinogenic effects, the FDA approved this dye only for limited applications such as coloring the peel of oranges, while in the EU it is not permitted at all [168]. However, there it was recommended that this dye should not be used as a food additive [184]. Therefore, this dye should not be incorporated into rodent diets.

Similarly, Acid Orange 137 (Orange B) is approved in the US for use in small traces (150 ppm) only in Frankfurter and sausage casings, while it is forbidden as a food additive in the EU [164,184]. Structurally, it is a pyrazolone dye that is reduced in the gut to form naphthionic acid [184]. In rodents this dye induces lymphoid atrophy of the spleen, bile-duct proliferation, and moderate chronic nephritis when applied for long-term [184]. Therefore, the usage of this dye as a food additive is forbidden in the EU [168].

In sum, the FDA has approved seven synthetic color additives for usage in food products. Three of them (Fast Green FCF, Citrus Red No. 2, and Orange B) are not permitted in the EU as food additives. Although the impact of artificial dyes on mice has not been intensively investigated, there are some studies showing that individual artificial dyes or combinations thereof might be neurotoxic when applied in high concentration [185]. However, typically the concentration necessary to induce adverse effects in male and female mice are extremely high. Brilliant blue FCF for examples showed no adverse effects even at high dietary concentration (7354 mg/kg/day and 8966 mg/kg/day) in male and female mice for 104 weeks [186]. These concentrations are far beyond the concentrations that are used for diet coloring of mice research diets.

## 11. Diversity of Diet Ingredients may Confound Data Interpretation

As discussed above, many papers using nutritional models in mice draw conclusions about dietary effects from comparison of grain-based diets with purified diets [145]. However, such mismatched diets potentially hamper the investigator’s ability to draw useful conclusions from otherwise well-designed studies [11]. The reproducibility of research findings is adversely affected by the use of improper control diets in metabolic disease research and the lack of adequate diet descriptions in resulting publications [11]. In many publications, grain-based diets referred vaguely as “chow diet”, “normal diet” or “control diet” are compared with purified ingredient diets also named as purified diets or semi-purified diets. Grain-based diets are made with grain, cereal ingredients, and animal by-products that may be somewhat variable from formulation to formulation, while purified diets are composed of highly refined ingredients [11]. Given these inherent differences between these diets, data produced from them should not be compared to each other or matched to one factor specifically different (e.g., fat or sugar composition). In particular, soluble fibers fermented by bacteria in the gut to SCFAs can for example change the gut pH, absorption of bile acids, and chelation of minerals [11]. Exemplarily, Chassaing and coworkers have demonstrated impressively that mismatched diets can result to erroneous conclusions [187]. In their study, the authors investigated the extent to which HFD-induced adiposity is driven by fat content vs. other factors that differentiate purified HFD, grain-based diet, and compositionally-defined diets (i.e., purified diets). Interestingly, the study revealed that high-fat content and lack of soluble fiber are both acting as obesogenic factors promoting rapid and marked loss of cecal and colonic mass and increased adiposity [187]. Therefore, the diet with its ingredients has to be considered as a key environmental factor that critically affects the outcome of a specific experiment. Properly matched “control diets” are therefore an indispensable prerequisite to draw conclusions regarding diet-driven phenotypic differences in respective studies.

## 12. Conclusions

Nutritional factors are crucial in laboratory animal science. To guarantee reproducibility of mouse experiments, it is necessary that they receive reliable food with constant composition. There are many providers that have concentrated on the production of grain-based and purified diets. Some are certified and produce their products according to national and international guidelines. The ingredients used are analyzed extensively and the production process guarantees nutrient stability and purity. Besides modification of the different ingredients of a diet, diets can be produced in varying shape, grains and colors using approved non-toxic food dyes. γ-rays and pasteurization are frequently used to sterilize diets fed in SPF facilities. However, these treatments might result in vitamin loss and formation of toxic substances, including acrylamide and peroxide radicals. An important but largely underestimated problem in conducting animal experimentation relying on nutritional models is the impact of confounding factors when choosing unsuitable control diets. These factors might impact the outcome of a specific experiment and incorrect conclusions. Confounding factors in this context are all ingredients differing between the control diet and the intervention diet. Likewise, the coloring with dyes such as Erythrosine and Tartrazine that have already shown to have biological effects in mice or humans should be ommitted for diet coloring. If an investigator has special requirements or wishes for his dietary interventions, it is urgently advisable to contact the manufacturer of the diet product before starting an animal experiment. In most cases, the manufacturers of diets offer consultations with expert nutritionists to assist the scientists in the selection of the right diet for the planned study requirement.

## Figures and Tables

**Figure 1 nutrients-12-00163-f001:**
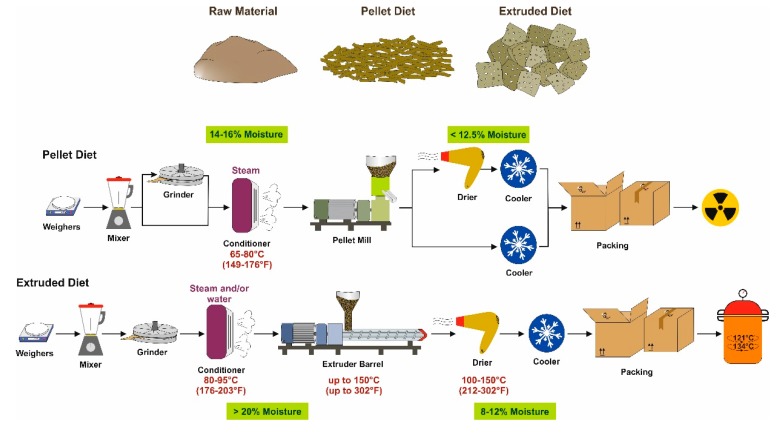
Schematic overview about the production process of mouse diets. Pellet and extruded diets are produced from the same raw materials. In a first step, the different ingredients are put together in the intended proportion, mixed, grinded to the desired density, and moistened to a desired moisture content (pellet diets: ~14–16%; extruded diet: >20%). Pellet diet is then processed in a pellet mill and dependent on the water content included either directly cooled or dried to a moisture content lower 12.5%. Thereafter, this diet is packed and sterilized for example by irradiation. In contrast, during the production of an extruded diet, the conditioned materials are then forced through an opening of a perforated plate or die to create a product in desired shape and size. It is then dried to moisture content of 8–12%, cooled, and packed. These diets are more or less germ-free because of the high temperature in the extruder barrel and drier. If necessary, these diets can be further sterilized by autoclaving before use. However, the high temperature during the extruding process already warrants low concentrations of microorganisms.

**Figure 2 nutrients-12-00163-f002:**
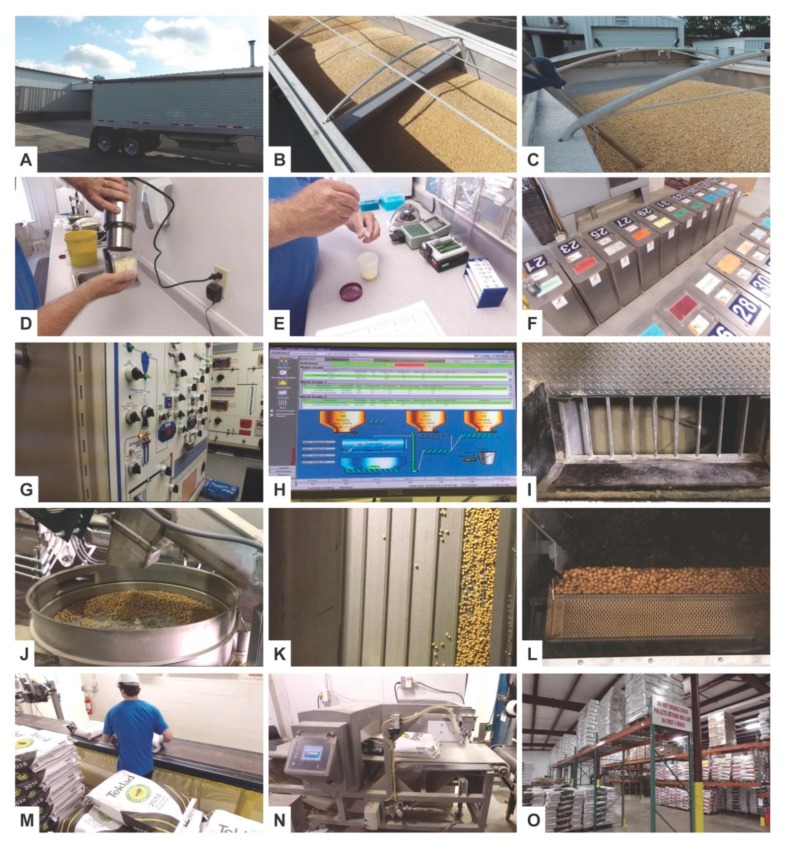
Overview of some work steps in the production of an extrusion diet in large scale. (**A**,**B**) The first step in production of mouse diets is the selection, procurement and approval of high quality bulk ingredients. In this regard, each company might have a different source of raw materials. (**C**–**E**) From each bulk, samples are taken and screened by a well-trained technician for mycotoxins (e.g., aflatoxin, vomitoxin, fumonisin), protein, fat, fiber, and moisture using near infrared spectroscopy and approved/certified test kits. (**F**) When quality is assured, each lot of raw material receives a lot number that is used to track each ingredient through the entire production process. (**G**–**I**) The different ingredients are mixed together in a certified mixer, in which flow from ingredient bins, scales and processing is critically monitored. (**J**–**L**) To produce an extruded diet form, the mixed ingredients are sent first to a post grind hammer mill and then to an extruder, in which the mixture is forced through a die. In this device, the product is expanded by a stream that is injected under pressure. In a next step, the product is passed through a dryer and several screeners to ensure that no metallic traces or other unwanted compounds are passed through. The moisture and bulk density of the product is evaluated and recorded. (**M**,**N**) Finally, the diet is packed in packing lines and once validated for unwanted stray metal particles by an in-line metal detection capability. (**O**) All diets are stocked in suitable facilities from which they are quickly retrievable on demand. All images were kindly provided by Dr. Jörg Lesting (Envigo Teklad Diets, Madison, WI, USA). A well-arranged movie showing the complete manufacturing process is viewable online at: https://www.envigo.com/p/teklad/.

**Figure 3 nutrients-12-00163-f003:**
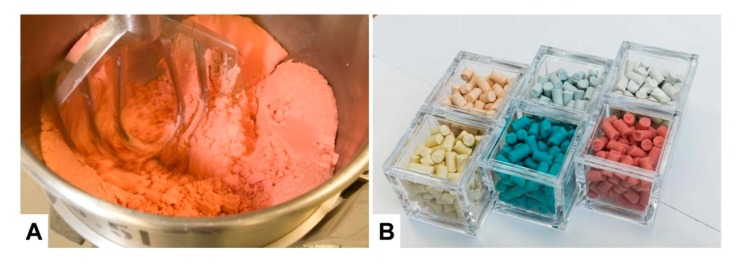
Production of customized purified mice research diets. (**A**) The manufacturing of a mouse diet used for basic and clinical research is a complex and highly-controlled process resembling that of producing bakery products for humans. Shown is a mixer in which the ingredients are blended. After that a known amount of water (based on the diet formula) is added and the product is properly shaped. The wet powdered diet is forced through a spinning die and without added heat, the pellets from their cylindrical shape, are cut to approximately the same length and then dried under low humidity conditions to remove the water added for pelleting. Dependent on the production process, different amount of diet ranging from several kilograms to several tons can be produced in a single batch. (**B**) Color coding by the addition of non-toxic dyes allow discriminating the different food products in animal facilities, in which animals are kept requiring different nutrients. The images depicted were kindly provided by Dr. Matthew Ricci (Research Diets, Inc., New Brunswick, NJ, USA).

**Figure 4 nutrients-12-00163-f004:**
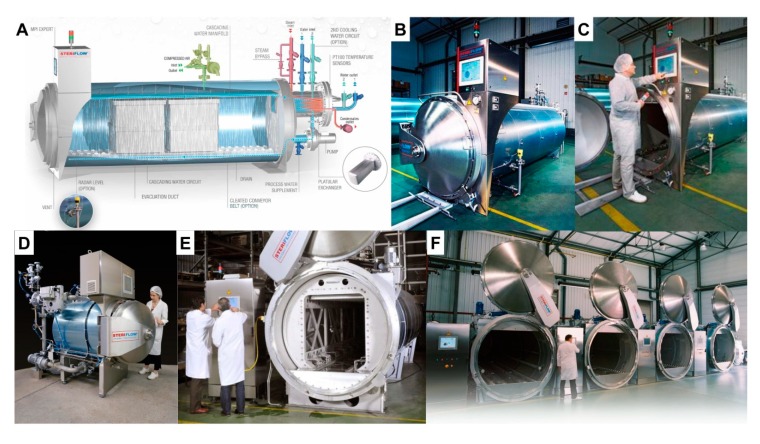
Sterilization of mouse diets by autoclaving. (**A**) Industrial autoclaves that can be used for sterilization of large batches of rodent diets must have large capacities. In principal, these devices are large pressure chambers, in which the goods are sterilized by subjecting them to a pressurized saturated steam at 121 °C (249 °F) maintained for about 20 min in a locked chamber. Different electronically-controlled valves and lines regulate steam flow and temperature in the steam chamber. (**B**,**C**) A typical industrial computer-controlled autoclave is depicted. (**D**–**F**) Autoclaves used for food production can be highly variable in size. Images showing different autoclaves from Steriflow (Roanne, France) were kindly provided by Kai Bergner from Vos Schott GmbH (Butzbach, Germany). A vivid 3D animation video of a typical water cascading process for sterilization by autoclaving can be found at: https://youtu.be/bWD87VVtzKU.

**Figure 5 nutrients-12-00163-f005:**
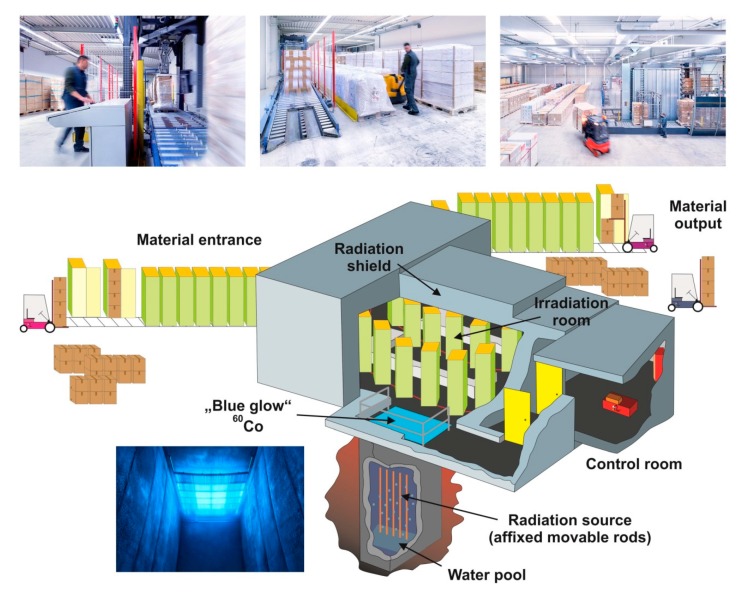
Irradiation of large batches of mouse diets. To minimize the risk of diet spoilage by pathogenic organisms, diets can be exposed to ionizing radiation. For irradiation in large scale, the packed diets are most commonly irradiated with γ-rays from a cobalt-60 (^60^Co) source that has high penetration depth and dose uniformity and is able to penetrate relatively dense products. When not in use, the radiation source is stored in a water-filled storage pool, which absorbs the radiation energy. This Cherenkov radiation results in a blue appearance of the water bath, which is commonly known as “blue glow”. For radiation, the ^60^Co rods are lifted out from this pool and the emitted energy is directed to the goods to be irradiated or the goods are moved around the γ-ray source. The facility is surrounded by a thick concrete wall to avoid radiation leakage into the environment. The photos of the irradiation facility and the blue glow were kindly provided BGS Beta-Gamma-Service GmbH & Co. KG (Wiehl, Germany, ©BGS/M. Steur).

**Figure 6 nutrients-12-00163-f006:**
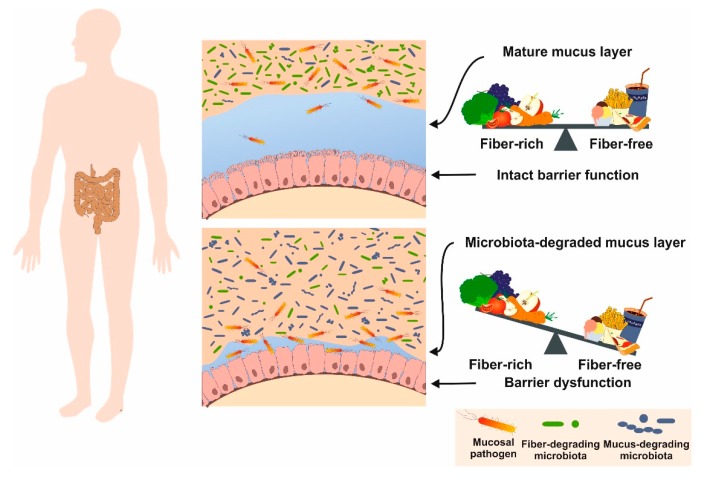
Beneficial effects of dietary fibers in humans. The ingestible parts of plants help to speed up the elimination of toxic waste in the digestive tract through promoting bowel movement in the colon and reacting with bacteria in the lower colon, thereby producing short chain fatty acids (acetate, propionate, butyrate) causing cancer cells to self-destruct. Inadequate fiber intake during malnutrition results in distortion of the mucosa, reduced intestinal barrier function, and inflammation. Similarly, fiber-free diets provoke degradation of mucus layer and barrier dysfunction in mouse.

**Figure 7 nutrients-12-00163-f007:**
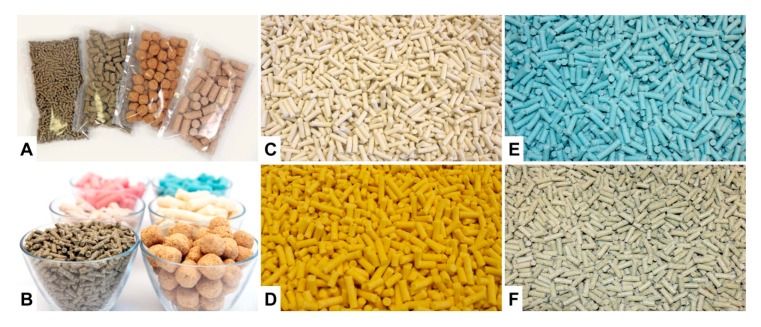
Diversity of mouse diets. (**A**) Diets can be produced in different appearance. (**B**) Diets can be produced in variable shape and color. (**C**) Uncolored diet, (**D**) yellow-stained diet, (**E**) blue stained high-fat diet, and (**F**) doxycycline hylate added diet. These figures were kindly provided by Dr. Dr. habil. Annette Schuhmacher (Ssniff Spezialdiäten GmbH, Soest, Germany).

**Figure 8 nutrients-12-00163-f008:**
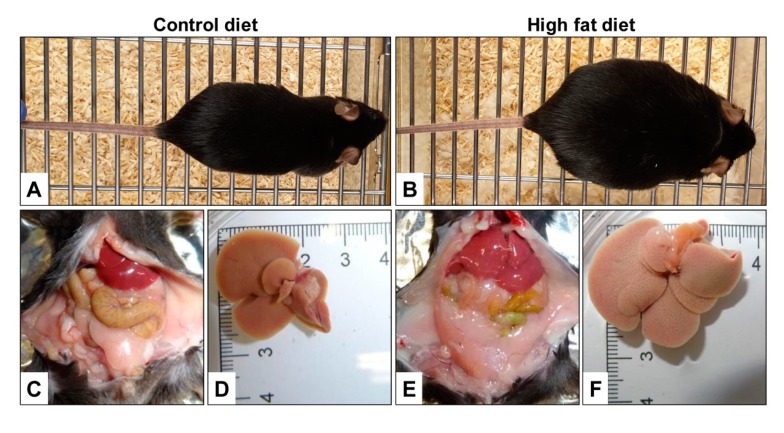
Diet-induced obesity and high-fat diets. (**A**,**B**) Comparison of mice receiving either a grain-based diet (**A**) or a diet enriched in fat (**B**) for prolonged times. While the body weight of the mice receiving a control diet was 25 g, the mouse fed a diet enriched for the same time was 52 g. (**C**–**F**) In obese animals, excess fat deposition and ectopic fat accumulation in the body occurs (**C**,**E**). In addition, the liver size is much higher in animals that received a diet rich in fat compared to animals at same age fed a grain-based diet (**D**,**F**). Depicted figures in (**A**,**B**) and (**C**–**F**) were kindly provided by Dr. Angela Schippers (Department of Pediatrics, UKA, Aachen, Germany) and Anastasia Asimakopoulou (IFMPEGKC, UKA, Aachen, Germany), respectively.

**Figure 9 nutrients-12-00163-f009:**
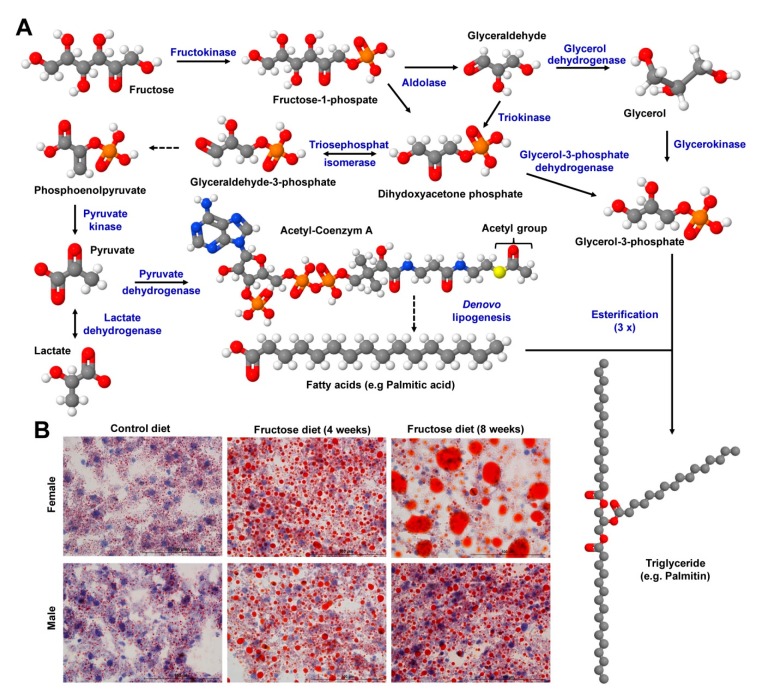
Fructose metabolism and consequences of increased fructose uptake. (**A**) The metabolism of fructose is initiated by phosphorylation of fructose to fructose-1-phosphate, which is subsequently hydrolyzed to form dihydroxyacetone phosphate and glyceraldehyde. Glyceraldehyde can also be converted to dihydroxyacetone phosphate or metabolized to glycerol 3-phosphate. Dihydroxyacetone phosphate can be isomerized via glycerol to glyceraldehyde 3-phosphate. Dihydroxyacetone phosphate can be further reduced to glycerol-3-phosphate or converted into glyceraldehyde 3-phosphate, and subsequently sequentially to phosphoenolpyruvate, pyruvate, and lactate. Pyruvate is central in feeding the citric acid cycle by transferring acetyl groups to coenzyme A, which is essential for the generation of fatty acids. Fatty acids can be esterified to glycerol-3-phosphate to generate triglycerides. Compound images were prepared with the Jmol program (www.jmol.org), version 14.2.15 using the compound identification (CID) nos. 5984, 65246, 751, 753, 1005, 754, 668, 754, 107735, 444493, 91435, 985, and 11147, respectively, (**B**) feeding of a fructose-enriched diet for 4–8 weeks results in progressive accumulation of hepatic fat in mice, which become evident in Oil Red O stain. Interestingly, the fat deposition is higher in female mice than in male littermates. More details about the biological effects and pathomechanism of fructose-induced fatty liver disease can be found elsewhere [46,47].

**Figure 10 nutrients-12-00163-f010:**
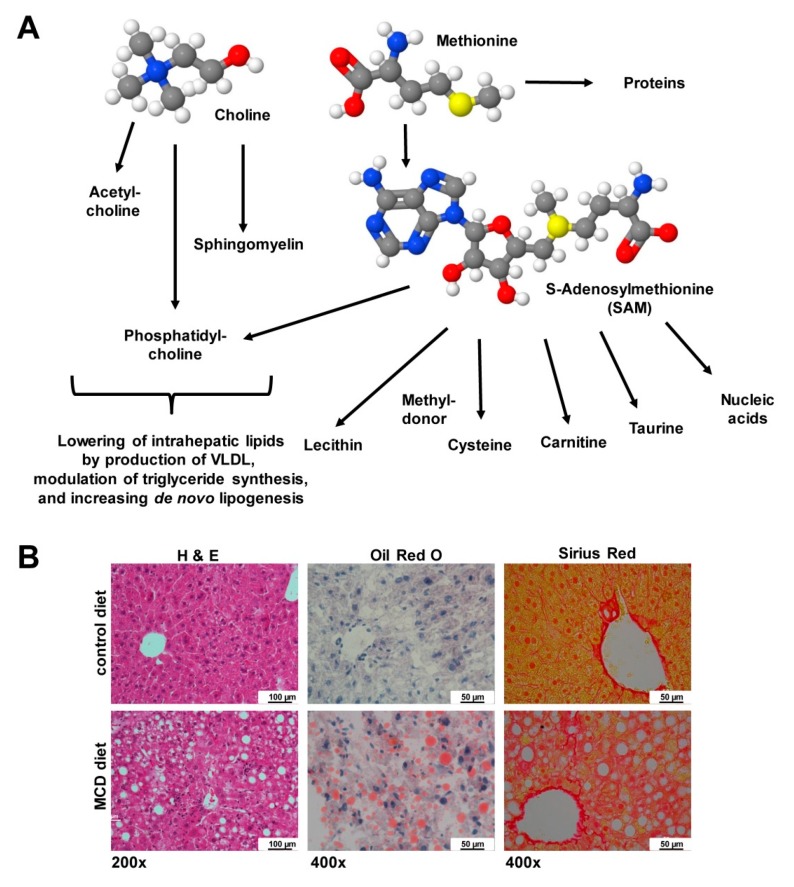
Choline and methionine are essential dietary supplements. (**A**) Methionine is an essential sulfur-containing amino acid that is part S-adenosylmethionine (SAM), which is indispensable as a methyl group donor in pathways driving synthesis of nucleic acids, proteins, lipids, and secondary metabolites. Choline is an integral part of phosphatidylcholines, sphingomyelins and necessary precursor for the synthesis of the neurotransmitter acetylcholine. Choline is necessary for production of very low density lipoproteins (VLDL), down-regulation of key enzymes involved in triglyceride synthesis, and proper function of de novo lipogenesis. Compound images were prepared with the Jmol program using the compound identification (CID) nos. 305, 6137, and 34755. (**B**) Mice fed a methionine-choline deficient (MCD) diet for four weeks develop severe hepatic liver damage, steatosis, ballooning, lobular inflammation, and fibrosis. In hematoxylin eosin (H & E) stain, the architectural changes are visible. In Oil Red O stain, the increased fat accumulation during the diet is assessable, while the Sirius Red stain is suitable to demonstrate increased deposition of collagens. Space bars correspond to 100 µm (H & E) or 50 µm (Oil Red O, Sirius Red). More details about the biological effects and pathomechanism of MCD diet-induced fatty liver disease can be found elsewhere [13].

**Figure 11 nutrients-12-00163-f011:**
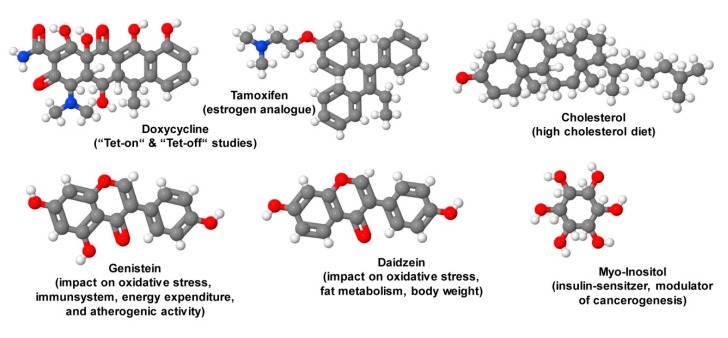
Some special ingredients in mouse diets used in biomedical research. Doxycycline, tamoxifen, genistein, daidzein, cholesterol, myo-inositol are compounds that are added to specific diets. Compound images were prepared with the Jmol program using the compound identification (CID) nos. 54671203, 2733526, 5280961, 5281708, 5997, and 892, respectively.

**Figure 12 nutrients-12-00163-f012:**
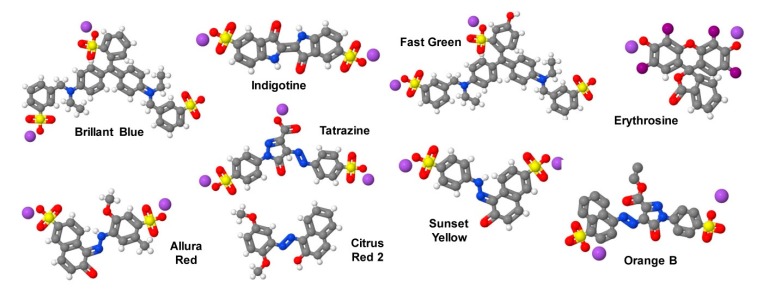
Artificial coloring of mouse diets. The artificial dyes Brilliant Blue, Indigotine, Fast Green, Erythrosine, Allura Red, Tatrazine, Citrus Red 2, Sunset Yellow, and Orange B or using their lakes has been permitted by the US Food and Drug Administration to color food products. These dyes are also approved for mouse diets. Compound images were prepared with the Jmol program using the compound identification (CID) nos. 19700, 2723854, 16887, 12961638, 33258, 164825, 22830, 17730, and 11685735, respectively.

**Table 1 nutrients-12-00163-t001:** Representative producers of mouse chows.

Company	Principal Office	Homepage
Research Diets, Inc.	New Brunswick, NJ, USA	https://www.researchdiets.com/
Ssniff Spezialdiäten GmbH	Soest, Germany	http://www.ssniff.com/
Altromin Spezialfutter GmbH & Co. KG	Lage, Germany	https://altromin.de/
BioServ	Flemington, NJ, USA	https://www.bio-serv.com/
Envigo Teklad (formerly Harlan Teklad)	Madison, WI, USA	https://www.envigo.com/
CLEA Japan, Inc.	Tokyo, Japan	https://www.clea-japan.com/
Specialty Feeds	Glen Forest, Western Australia, Australia	http://www.specialtyfeeds.com
Safe	Augy, France	www.safe-diets.com/
LabDiet	St. Louis, MO, USA	https://www.labdiet.com/
TestDiet	St. Louis, MO, USA	https://www.testdiet.com
Dyets, Inc.	Bethlehem, PA, USA	https://dyets.com/
Special Diets Services (SDS)	London, UK	http://www.sdsdiets.com/

**Table 2 nutrients-12-00163-t002:** Typical purified ingredients for special nutritional requirements of diets.

Substance Class	Representative Compounds
Proteins	Casein, soy protein isolate, egg white protein; crystalline amino acids
Carbohydrates	Sucrose, fructose, corn starch, (nondigestable oligosaccharides)
Fats	Lard, corn oil, safflower oil, Menhaden oil; soybean oil
Vitamins	Vitamin A, vitamin D, vitamin E, vitamin K, vitamin B_12_, biotin, choline, folates, niacin, pantothenic acid, vitamin B_6_ (pyridoxine, pyridoxal, pyridoxamine), riboflavin, thiamine, vitamin C
Minerals	Dicalcium phosphate, sodium selenite
Fibers	Cellulose, guar gum, pectin, carboxymethylcellulose, carrageenan, xanthan gum, gum arabic, inulin, fructooligosaccharides
Additives	Genistein, daidzein, cholesterol, myo-inositol
Special additives	Doxycycline, tamoxifen

**Table 3 nutrients-12-00163-t003:** Biochemical function of water-soluble and fat-soluble vitamins.

Vitamin Sub-Class	Vitamin	Compound (Alternate Names)/Members	Biochemical Function	Reference
Water-soluble	Vitamin C	Ascorbic acid	Maintenance of redox balance; co-substrate for several enzymes; intracellular antioxidant; electron donor	[64]
	Vitamin B_1_	Thiamine	Coenzyme in the catabolism of sugars and amino acids	[65]
	Vitamin B_2_	Riboflavin	Coenzyme in flavoprotein enzyme reactions (e.g., FAD); antioxidant	[66]
	Vitamin B_3_	Niacin (nicotinic acid)	Coenzyme involved in protein, fat and carbohydrate metabolism (e.g., Nicotinamide adenine dinucleotide (NAD); hydrogen carrier; antioxidant, reducing agent	[67]
	Vitamin B_5_	Pantothenic acid	Coenzyme in synthesis and metabolism of proteins, carbohydrates and fats; required for synthesis of coenzyme A	[68]
	Vitamin B_6_	pyridoxine, pyridoxal, pyridoxamine and their respective mono-phosphorylated derivatives	Coenzyme in many enzymatic reactions (decarboxylations, transaminations, eliminations, racemizations, transsulfurations, interconversions); antioxidant	[69]
	Vitamin B_7_	Biotin (vitamin H)	Coenzyme for carboxylases: pyruvate carboxylase, 3-methylcrotonyl-CoA carboxylase, propionyl-CoA carboxylase, and coenzyme for acetyl-CoA carboxylase 1 and 2	[70]
	Vitamin B_9_	Folic acid (folacin), folate pteroyl-L-glutamic acid, pteroyl-L-glutamate, pteroylmonoglutamic acid	Coenyzme in single-carbon group (methyl-, methylene-, formyl group) transfer reactions	[71]
	Vitamin B_12_	Cobalamin	Coenzyme for the methionine synthase and methylmalonyl-CoA mutase	[72]
Fat-soluble	Vitamin A	Retinol, retinal, retinoic acid, provitamin A carotenoids	Modulator of immune system; low-light and color vision; wound healing; hormone (binding to retinoic acid receptors); metabolic effects; reproduction	[73,74]
	Vitamin D	Group of secosteroids (e.g., ergocalciferol, cholecalciferol and others);	Binding to vitamin D receptor acting as a transcription factor; calcium and phosphate homeostasis; immune system; cell proliferation and differentiation; bone formation; innate and adaptive immunity	[75,76]
	Vitamin E	α/β/γ/δ-tocopherols, α/β/γ/δ-tocotrienols	Antioxidant and radical scavenger; modulator of gene expression; enzyme activity regulator (e.g., protein kinase C)	[77]
	Vitamin K	Phylloquinone (vitamin K1), menaquinones MK-4 through MK-10 (vitamin K2)	γ-glutamyl carboxylation; relevant in blood coagulation and bone metabolism; modulator of transcriptional activity; agonist of steroid and xenobiotic nuclear receptor; neural stem cell differentiation modulator	[78]

**Table 4 nutrients-12-00163-t004:** Vitamin losses during pelleting, extrusion and storage of feeds and factors affecting vitamin deterioration.

Vitamin	Factors Affecting Vitamin Stability during Processing and Storage
Vitamin C (ascorbic acid)	Moisture, heat, oxidation, light, iron
Vitamin B_1_ (thiamine)	Oxidation
Vitamin B_2_ (riboflavin)	Light
Vitamin B_3_ (niacin)	Rather stable
Vitamin B_5_ (pantothenate)	Oxidation, light
Vitamin B_6_ (pyridoxine)	Oxidation, reduction
Vitamin B_7_ (biotin)	Oxidation
Vitamin B_9_ (folic acid)	Oxidation, light, microbial
Vitamin B_12_ (cobalamin)	UV light, interaction with other water-soluble vitamins, heat, pH
Vitamin A (retinol, retinal, retinoic acid, provitamin A carotenoids)	Oxidation, light, (trace elements)
Vitamin D (cholecalciferol)	Stable
Vitamin E (α/β/γ/δ-tocopherols, α/β/γ/δ-tocotrienols)	Oxidation, light, oxidized fat
Vitamin K (menadione)	Oxidation

**Table 5 nutrients-12-00163-t005:** Nutrient requirements of mice maintained in conventional animal facilities *.

Nutrient	Amount (per kg Diet)
Lipid	50 g
Linoleic acid	6.8 g
Protein (N × 6.25) **	180–200 g
Amino acids	Arginine: 3 g; histidine: 2 g; isoleucine: 4 g; leucine: 7 g; valine: 5 g; threonine: 4 g; lysine: 4 g; methionine: 5 g; phenylalanine: 7.6 g; tryptophan: 1 g
Minerals	Calcium: 5 g; chloride: 0.5 g; magnesium: 0.5 g; phosphorus: 3 g; potassium: 2 g; sodium: 0.5 g; copper: 6 mg; iron: 35 mg; manganese: 10 mg; zinc: 10 mg; iodine: 150 µg; molybdenum: 150 µg; selenium: 150 µg
Vitamins	Retinol: 0.72 mg (= 2400 IU); cholecalciferol: 0.025 mg (1000 IU); RRR-α-tocopherol: 22 mg (= 32 IU); phylloquionone: 1 mg; biotin: 0.1 mg; choline: 2 g; folic acid: 0.5 mg; niacin: 15 mg; Ca-pantothenate: 16 mg; riboflavin: 7 mg; thiamine-HCl: 5 mg; pyridoxine-HCl: 8 mg; cobalamin: 10 µg

* The information depicted for individual nutrients was taken from the National Research Council (NRC) guidelines [10]. According to these guidelines, the nutrient requirements are expressed on an as-fed basis for diets containing 10% moisture and 16–17 kJ metabolizable energy per g and should be adjusted for diets of differing moisture and energy concentration. ** This calculation of this parameter assumes that the average nitrogen (N) content of proteins is about 16 percent, which led to the use of the calculation N × 6.25 (1/0.16 = 6.25) to convert nitrogen content into protein content. The amount of protein is given for animals maintained under regular growth conditions.

**Table 6 nutrients-12-00163-t006:** Consequences of insufficient supply with selected nutritional components.

Component	Consequences of Insufficient Supply	References
Fat	Alterations in composition and functionality of synaptosomal plasma membranes	[93]
Protein	Decreased host immune defense; reduced numbers of splenocytes, lower quantities of glutathione in several organs; less food ingestion and weight gain during pregnancy resulting in fewer viable pups	[94,95]
Phe, Thr, Trp, Met, Lys, Leu, Ile, Val	Massive reduction of abdominal fat mass after 7 days, most likely via increased energy expenditure	[96,97,98,99]
Calcium	Development of osteoporosis-like symptoms with reduced femur length and reduced density of various bones	[100]
Magnesium	Hypomagnesemia reduced bone growth and chondrocyte functionality; During embryogenesis embryotoxic effects (retardation, disturbed bone development and skeletal malformations) are induced	[101,102]
Phosphorus	Severe growth retardation with a 50% reduction in body weight; reduced formation of milk droplets	[103,104]
Potassium	Hypokalemia results in decrease in luteinizing hormone and testosterone provoking testicular impairments that is also seen in a fall in the weight of seminal vesicles	[105]
Copper	Development of anemia, duodenal hypoxia and alterations in intestinal iron absorption	[106]
Iron	Development of pronounced splenomegaly; anemia with reduction in hemoglobin and hematocrit levels with higher risk of developing deep dental caries	[107,108]
Manganese	Growth retardation and malfunction of reproductive organs and ovulation; congenital debility of young animals with loss of both equilibrium and coordination	[109,110]
Zinc	Reduced immune responses after challenging with pathogen indicated by greater weight loss, stool shedding, mucus production and diarrhea	[111]
Iodine	Development of carcinoma of the thyroid; formation of oxidative stress and DNA modifications	[112,113]
Selenium	Potentiate the development of autoantibodies; reduction of amino acid levels and elevation of mononucleotides resulting in dysregulated metabolomes and age-associated decline of protein synthesis; development of widespread pyogranulomas	[114,115,116]
Thiamine	Reduction of energy state in the liver; reduction of blood glucose, insulin, triglycerides, cholesterol, liver glycogen; increase of serum lactate	[117]
Biotin	Impairment of mitochondrial structure and function; intoxication with propionyl-CoA; systemic inflammation	[118]
Vitamin D	Increased expression of sex steroid receptors in myometrium; increased expression of proliferation-related genes; promotion of fibrosis, inflammation, and immunosuppression; enhancement of DNA damage; increased lipid deposition in skeletal muscle and muscle fiber disorganization	[119,120]
Vitamin A	Breakdown of oral tolerance; reduction of iron absorption	[121,122]
Linoleic acid	Reduction results in lower concentrations of circulating, small bowel and hepatic endocannabinoids; lower feed efficiency and weight	[123]
Choline	Amplification of liver fat accumulation in phases of high fat consumption; lowering of fasting plasma insulin; improvement of glucose tolerance; reduction of fibroblast-like cells in circulating tumor cells and less metastasis	[124,125]

**Table 7 nutrients-12-00163-t007:** Composition of representative mouse diets used in hepatology research *.

	Component	Control Diet	DIO ** Diet	WD Diet	FFC Diet	MCD Diet
**Crude nutrients** **[%]**	Crude protein (N × 6.25)	17.6	24.4	17.3	19.7	15.0
	Crude fat	7.1	**34.6**	**21.1**	20.0	10.0
	Crude fibre	5.0	6.0	5.0	5.1	3.0
	Crude ash	3.2	5.3	4.2	4.4	3.3
	Starch	38.2	0.1	14.4	0.1	19.2
	Dextrin	**13.1**	**15.4**	0	**10.9**	0
	Sugar	11.2	9.4	34.3	34.2	45.1
	N free extracts	63.1	26.3	49.8	46.2	67.3
**Minerals** **[%]**	Calcium	0.55	0.92	0.76	0.78	0.59
	Phosphorus	0.37	0.64	0.45	0.50	0.46
	Sodium	0.15	0.20	0.24	0.20	0.16
	Magnesium	0.10	0.23	0.10	0.13	0.06
	Potassium	0.55	0.97	0.54	0.77	0.36
**Fatty acids** **[%]**	C 4:0	-	-	**0.80**	-	-
	C 6:0	-	-	**0.53**	-	-
	C 8:0	-	-	**0.29**	-	-
	C10:0	-	-	**0.63**	-	-
	C 12:0	-	**0.07**	**0.72**	0.02	-
	C 14:0	0.02	0.44	2.22	0.07	-
	C 16:0	0.84	7.93	5.60	2.04	1.09
	C 17:0	0.01	0	0.14	0.01	0.01
	C 18:0	0.26	4.37	2.05	2.38	0.19
	C 20:0	0.03	0.11	0.04	0.14	0.03
	C 16:1	0.01	0.94	0.38	0.08	0.01
	C 18:1	1.78	13.97	4.65	11.65	2.58
	C 18:2	3.69	4.64	0.38	2.14	5.53
	C 18:3	0.41	0.49	0.11	0.19	0.09
**Amino acids** **[%]**	Lysine	1.47	2.20	1.43	1.63	1.40
	Methionine	0.55	0.86	0.93	0.61	**0**
	Cystine	0.37	0.45	0.07	0.43	0.35
	Threonine	0.78	1.07	0.76	0.86	0.80
	Tryptophan	0.23	0.33	0.22	0.26	0.18
	Arginine	0.69	0.95	0.67	0.77	0.98
	Histidine	0.53	0.74	0.52	0.59	0.45
	Valine	1.23	1.70	1.20	1.37	0.81
	Isoleucine	1.00	1.38	0.97	1.11	0.80
	Leucine	1.75	2.42	1.71	1.95	1.10
	Phenylalanine	0.92	1.27	0.89	1.02	0.74
	Phenylalanine +Tyrosine	1.85	2.56	1.80	2.06	1.24
	Glycine	0.35	0.52	0.34	0.39	2.31
	Glutamic acid (+Glutamine)	3.97	5.50	3.88	4.43	3.96
	Aspartic acid (+Asparagine)	1.31	1.82	1.28	1.45	0.95
	Proline	2.02	2.80	1.97	2.25	0.35
	Serine	1.06	1.46	1.03	1.18	0.35
	Alanine	0.53	0.81	0.52	0.59	0.35
**Vitamins** **[mg per kg]**	Vitamin A ***	1.20	4.50	4.50	1.32	5.85
	Vitamin D_3_ ****	0.025	0.0375	0.0375	0.0275	0.055
	Vitamin E	75	150	150	90	135
	Vitamin K (as MNB)	4	20	20	4	45
	Thiamine (B_1_)	12	25	26	13	22
	Riboflavin (B_2_)	16	16	16	18	22
	Pyridoxine (B_6_)	7	16	16	7	22
	Cobalamin (B_12_)	0.025	0.03	0.03	0.028	0.03
	Nicotinic acid	29	47	49	32	98
	Pantothenic acid	15	55	55	17	60
	Folic acid	2	16	16	2	2
	Biotin	0.2	0.3	0.3	0.2	0.4
	Choline	1130	1140	920	920	**0**
**Trace elements** **[mg per kg]**	Iron	49	168	49	68	42
	Manganese	22	95	22	30	53
	Zinc	41	65	41	58	29
	Copper	10	13	11	14	6
	Iodine	0.3	1.2	0.3	0.4	0.2
	Selenium	0.2	0.2	0.2	0.2	0.1
**Other ingredients** **[mg/kg]**	Cholesterol	0	**~230**	**2070**	**20,000**	0

* The concentration of the individual components were taken from Ssniff diets with order numbers E15712 (control), E15742 (DIO), E15721 (Western diet), E15766-3402 (NASH diet), and E15653 (MCD diet), respectively. ** Abbreviations used are: DIO, diet-induced obesity; FFC, fat-, fructose- and cholesterol-rich diet; HF, high fructose; MCD, methionine-choline deficient; MNB, menadione nicotinamide bisulfite; WD, Western diet. *** 1 mg Vitamin A corresponds to 3333 IU. **** 1 µg Vitamin D_3_ corresponds to 40 IU. Percentages [%] are given in relation to the whole weight of the diet. Underlined values mark special features in the respective diet.

**Table 8 nutrients-12-00163-t008:** Dyes allowed for artificial coloring in the US.

Food and Drug Administration (FDA) Name	Organic Compound	Color	E Number	ADI * (mg/kg bw/d)	Chemical Abstracts Service (CAS) Number
Federal Food, Drug and Cosmetic Act (FD&C) Blue No. 1	Brilliant Blue FCF (Acid Blue 9, D&C Blue No. 4, Atracid Blue FG)	blue	E133	0–6	3844-45-9
FD&C Blue No. 2	Indigotine (Indigo Carmine, Acid Blue 74, Murabba, Sachsischblau)	indigo	E132	0–5	860-22-0
FD&C Green No. 3	Fast Green FCF (Food Green FCF, Food Green 3, Green 1724)	green	E143 ***	0–12.5	2353-45-9
FD&C Red No. 3	Erythrosine (Erythrosin B, Acid Red 51, Pyrosin B, Food Red No. 3)	pink	E127	0–0.1	16423-68-0
FD&C Red No. 40	Allura Red AC (Food Red 17, Curry Red)	red	E129	0–7	25956-17-6
FD&C Yellow No. 5	Tartrazine (Tatrazol Yellow, Acid Yellow 23, Food Yellow 4)	yellow	E102	0–10	1934-21-0
FD&C Yellow No. 6	Sunset Yellow FCF (Orange Yellow S, Food Yellow 3)	orange	E110	0–4	2783-94-0
Citrus Red 2 **	Citrus Red (2,5-Dimethoxy-1-Phenylazo-2-naphthol, CI 12156, CI Solvent Red 80)	red	E121 ***	Only approved for use to color orange peels (US)	6358-53-8
Orange B **	Acid Orange 137 (LS-128771, Schembl132534)	orange	not allowed ***	Only approved for use in hot dog and sausage casings (US)	53060-70-1

* Acceptable daily intake (ADI) values are given for humans and were taken from the Joint FAO/WHO Expert Committee on Food Additives (JECFA, https://www.who.int/foodsafety/areas_work/chemical-risks/jecfa/en/) or from the Internationally Peer Reviewed Chemical Safety Information (http://www.inchem.org). Abbreviations used are: bw, body weight; ** Dye allowed by the FDA for limited applications. *** The usage of this dye in food products is forbidden in the EU.

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
