# Peer review of "All You Can Feed: Some Comments on Production of Mouse Diets Used in Biomedical Research with Special Emphasis on Non-Alcoholic Fatty Liver Disease Research"

_nutrients, 2020, doi:10.3390/nu12010163_

Round 1

Reviewer 1 Report

Brief Summary:

This review is to discuss mouse diets used in biomedical research.  The topics range from describing different ingredients used in diets, some descriptions of general diet production methods, nutrient requirements, effect of nutrients on health, certain diets for diet-induced diseases, colorants in diets, and sterilization techniques. 

Broad Comments:

This review is titled in a very open ended manner, which is appropriate because this is a very open ended review and in my opinion, too open ended.  There should be a goal in mind that is more specific.  Instead, this review is just a very random collection of information, with no flow, and also contains a lot of unnecessary details.  The history of the mouse in research is appropriate, but could have been shortened and they also provide random bits of information on different ingredients. There is also a good bit of time spent on diet-induced phenotypes.  Again, what is the goal of the review?  For a review that is supposedly to describe some facts about mouse diets, they don’t really provide a methodical review of the general types of mouse diets (i.e. grain-based vs. purified diets), which should be done first and then go into what they want to discuss about the diets. 

For a review that is to talk about mouse diets, they should define well what the differences are between purified diets and grain-based diets and stick with those terms without deviating.  They use several terms when discussing diets throughout the review and they need to be more consistent.  For example, they use ‘mice diets’, ‘rodent chows’, ‘custom diets’, ‘defined diets’ ‘purified chows’, ‘special diet’, and ‘purified ingredient diets’, ‘grain-based diets’ and ‘standard chows’.  There are 2 basic diet types available to researchers and I feel they don’t do a good job in defining these right in the beginning – in fact, the paragraph right before the conclusion does the best job defining the 2 diet types – grain-based diets and purified ingredient diets.  

There is a lot of discussion of human data throughout the review, which is not relevant to the topic of mouse diets. There is a section devoted all to different colorants and I feel this is to a large degree unnecessary and with a lot of it describing how these colorants influence humans, which really isn’t relevant. 

They also provide information on certain diet-induced phenotypes.  Again, it is unclear why – if this paper is to review phenotypes, then this should be in the title.  They also seem to put some more focus on diet-induced liver phenotypes, but this paper will not be as helpful as others that have been published, so rather than going into much detail, they can mention and provide examples of diets for driving metabolic diseases including NAFLD, but no need to go into too much detail in terms of mechanisms, etc. 

More specific comments:

The subtitles in the paper many times are vague or perhaps inappropriate.  For example, one called ‘Producers of mice diets’ provides some information on care of certain mice and certain agencies that provide guidelines and regulations for manufacturers.  Rather, they should title this ‘producers of mouse diets’.

They mention that each diet formula is manufactured by a fixed formula designed by a nutritionist.  Not always – there are certain diet manufacturers that use ‘constant nutrition’ which will modify the amount of an ingredient if a given batch of an ingredient contains different levels of a nutrient than the previous batch.  Also, the fact that a fixed grain-based diet is fixed doesn’t guarantee the consistency of the nutrient profile. 

In the section titled ‘production process’, they do describe production of some diets, like fixed formulas and Open Source diets, which should actually be renamed as purified diets because the term Open Source is really branding.  They then go into differences between ‘standard chows’ and ‘purified chows’ – they should rename ‘standard chows’ and call them cereal-based or grain-based diets and purified diets – the term ‘chow’ should be replaced with ‘diet’.  Grain-based or cereal-based diet is more descriptive than ‘standard chow’.  They should also mention some of the first purified diet formulas, the AIN diets. 

Table 1 should be limited to producers of the diets rather than suppliers.  There are many other suppliers than what are listed.

Table 2 needs revision – many of the fibers listed are not purified ingredients – in fact, the only ones that are purified are cellulose, guar gum, pectin, carboxymethylcellulose, carrageenan, xanthan gum, and gum Arabic.  Inulin and fructooligosaccharides could be add to this list. All other fibers would be found in grain-based diets.  Also, some fibers listed don’t exist as a lone fiber source, like hemicellulose and lignins, both of which are fiber types again found in grain-based diets. 

There are many tables and figures that are not necessary and also presented not so well. The title of table 3 is not really correct because there is just a list of vitamins and their biochemical functions rather than consequences of nutrient insufficiencies.  I feel this table is not necessary and they should just refer to a good reference for this information.  Also, vitamin C is not required for the mouse.

Tables 4 and 5 are ok given this pertains directly to mouse diets. Table 6 is not necessary and really could just be referenced unless the point of this paper is to describe diet-induced phenotypes. It would be better to perhaps just say that these diets can be made with varying levels of vitamins and minerals to address diet-induced phenotypes and perhaps point out some of the phenotypes in the text.  I would therefore remove these tables of information

In the protein section (line 250), they bring up egg white as a protein source, but this is not a source that is used often – typically only used for zinc deficient diets as casein contains more zinc than egg white.  They forget to mention that additional biotin is needed when using egg white due to the presence of avidin which can reduce biotin bioavailability.  They also could mention crystalline amino acids as an alternative to complete proteins.

They then decide to choose 4 different fats in purified diets, very randomly – there are many other fats.  One they should have included is soybean oil which is contained in AIN-93 series diets and many high fat diets as a source of essential fatty acids.  It is strange they include safflower oil and menhaden oil, both of which are not often used.  For corn oil, they don’t mention why it is commonly added to diets and that it was the fat used in the AIN-76A diet. 

Discussing the stability of the vitamins is appropriate.

The discussion of fiber (line 394-407) is not very clear.  Fiber dilutes the nutrient and energy density, but they make it sound like that’s the reason it is added to diets.  In fact, they are added intentionally in purified diets to provide health to the gut and found inherently in ingredients of grain-based diets.  They also don’t define crude fiber well as they consider it as a subclassification of NDF and ADF (?), and really it is a measure that encompasses mainly cellulose and lignin (ADF) but not any soluble material.  The total of hemicellulose, cellulose and lignin (NDF) is not even total fiber – the method that is done to determine this misses some of the soluble fiber, pectin.  There is also some other descriptions that are not clear, such as “ADF is the fraction of fibers that can be used by the animal to extract energy, while NDF is a measurement for the total contents of fiber that an animal take up” – no energy is extracted from cellulose and lignin and what do they mean by ‘take up’?  They don’t mention anything about soluble vs insoluble fiber – when taking about fiber being fermented, it should be made clear that not all fibers are fermented.  They are obviously not fiber experts, but a little more reading on this topic is needed before they talk about this topic.  The figure 4 should be removed or made more clear.  They include human foods and a picture of a human when this is for mouse work – again, not relevant to the topic.

They then talk about nutrient requirements of mice, which I can see to some extent, but they can just reference and not include the table 5. 

Regarding Table 6, this is a very large table of information that I’m not sure is relevant because it is not a major topic of discussing in the review.  The authors could simply reference to the nutrient requirements rather than including all this information here.  If this review were to be a resource for consequences of nutrient deficiencies, it would be relevant. 

The subtitle ‘rodent chows’ (line 458) is really vague.  They start talking about gut microbiota and how it along with overfeeding or fasting can cause disease – how long are they fasting?  Then they talk about caloric restriction.  Then they talk about the mouse, then they talk about contaminants – unsure of the path of this topic. 

In line 477, they turn to diet-induced disease and provide a table (7) with compositions of mouse diets in hepatology research.  If they are to do this, they really should provide formulas – tell us the protein, fat, and fiber sources rather than a chemical composition.  For example, what is the protein source (casein, amino acid?), fat source (lard, soybean oil?), fiber source?  These formulas being purified should be open to the researcher and the reader.

In line 511, they mention the four diets (in the table) typically induce obesity – MCD diets (one of the diets) do not induce obesity, insulin resistance and hypertension, which it doesn’t.  The other diets, DIO, WD, and FFC will drive some of these phenotypes, but not hypertension (only if sodium chloride is added).

In line 513, I was confused how they described NAFLD as the progressive form characterized by obesity, diabetes, insulin resistance, and hypertension.  NAFLD refers to just the liver disease component rather than the other diseases – rather, they should call this collection of diseases (i.e. obesity, diabetes, insulin resistance, hypertension, and NAFLD) as metabolic disease. 

In line 530, they describe western diets as characterized by high content of fat combined with high amount of sugar.  They mention these diets are enriched with traces of SCFAs, which is thought to increase ghrelin because it requires caprylic (misspeleed as cyprylic).  However, it sounds like this doesn’t impact body weight or food consumption, so I’m confused why this idea was brought up.

Line 627, they talk about ‘special ingredients’ including tamoxifen, cholesterol, genistein, daidzein, and myo-inositol.  I’m not sure why they chose these 5 random ingredients.   

Line 697, They then discuss food coloring for a long time, mentioning several colors that are not being used in lab animal diets and how they affect humans mainly.   Too much time is spent talking about how the dyes influence humans.  There is no reason to bring up all these food colorants such as Blue #2, Green #3, Orange B.  they also mention that the concentrations studied typically are much beyond what are typically added to diets.  Therefore, it seems of little value to the reader to describe these in such detail when they are not included in very high levels in mouse diets anyway.

Discussion of autoclaving and irradiation is fine.

In the end paragraph, they discuss ‘diversity of diet ingredients and confound data interpretation’ – This makes sense to raise in a review about mouse diets.  The discussion of ingredients in grain-based diets vs. purified diets should be more detailed.  However, as stated previously, they should have defined the differences between grain-based diets and purified diets much better in the beginning of the review.

Overall / merit

In the end, I am wondering whether this review is covering too many random details.  They should refine it better and focus more on defining laboratory animal diets and their components.  There are many details that are unnecessary and it seems that the work is being padded by unnecessary details.  There are too many very vague subtitles that should be defined better and stay within a given topic.  There is also too much discussion of human data and the authors also don’t need so much information on the history of the mouse in the beginning.  After all, this is about mouse diets, so if there should be a history lesson, mouse diets should be the focus, not the mouse itself. Extensive revisions are required.  They need to first define what they want to focus on – is it more on the diet-induced phenotypes or will it be more on the production of the diets?  These are 2 very different paths.  If they’re focusing on certain ingredients, there should be more rationale of why to focus on certain ingredients.

English level

There are many grammatical issues.  ‘Mice diets’ should be corrected to ‘mouse diets’.  In addition, several sentences are not well structured and require modification.

Author Response

Dear reviewer,

please find our response to your concerns/suggestions in the pdf-file attached.

Reviewer 2 Report

The authors have written a comprehensive review on diets fed to laboratory grown mice. The review is very detailed and well organized. The authors have included relevant points and have discussed caveats when applicable. 

Can the authors discuss little about where the manufacturers get the raw materials of diet? I suppose that is also regulated and is similar between manufacturers to minimize differences? when mice move from from institute to another, do their regular diet change significantly? or do all institutes have similar diets? Can the authors discuss this? Do researchers that study immune system in the gut or gut microbes, use a specific kind of diet? The authors have discussed various kinds of diets. It will be good if they discuss this one too. The authors can move the "pasteurization" and irradiation sections (lines 823-899) earlier in the text (after section 3: the production process). 

Author Response

Dear reviewers,

please find our response to your comments/suggestions in the pdf-file attached.

Reviewer 3 Report

Weiskirchen et al. prepared the review that draws attention to the importance of mouse diets in the field of biomedical research. The article is well written in terms of the format, inclusion of up-to-date information. The article is very informative to many researchers using experimental animals. I have only minor comments as listed below.

<Minor comments>

Line 52, and line 499. ‘Hepatology’ would better be changed to ‘hepatology’.

Line 60. ‘27 Member States’ would better be changed to ‘27 member states’.

Line 202. High fat diet.

‘high fat diet’, ‘high-fat diet’, and ‘HFD (in line 916, page 30. without spelling out) are all used in the manuscript, tables, and figure legends. They should be unified.

Line 207. APOE4 deficient mice

This sentence is incorrect. ‘APOE4 deficient mice’ should be ‘APOE4 transgenic mice’

Please note that ApoE has .4 polymorphic alleles and APOE4 is one of them.

Line 253. ‘distort’ would be changed to ‘distorts’.

Line 305. ‘fatty tissue’ would be changed to ‘fat tissue’ or ‘adipose tissue’.

Line 335. ‘Menhaden’ would be changed to ‘menhaden’.

Line 498. ‘type II diabetes’ would be changed to ‘type 2 diabetes mellitus’.

Lines 498-499. ‘liver damage’ would be changed to ‘liver injury’.

Line 501. ‘a diet enrich in fat,,,,’ would be changed to ‘a diet rich in fat,,,,’.

Line 511. ‘diabetes type II’ would be changed to ‘type 2 diabetes mellitus’.

Line 527. ‘mouse’ would be changed to ‘mice’.

Line 532. ‘Medium-chain fatty acids’ would be changed to ‘medium-chain fatty acids’.

Line 539. ‘a DIO’ would be changed to ‘DIO diet’ or ‘HFD’.

Line 555. ‘denovo’ would be changed to ‘de novo’.

Line 585 ‘choline-deficient (MCD) diet’ would be changed to ‘MCD’, since it first appears in line 502.

Line 622. ‘several’ would be changed to ‘twelve’, since the year 2008 is more than a decade ago.

Line 717. ‘Table 6, Figure 9’ would be changed to ‘Table 6, Figure 10’.

Line 835. ‘Figure 10’ would be changed to ‘Figure 11’.

Lines 916-7. ‘HFD’ would be spelled out as ‘high fat diet’. Alternatively, please consider to add the abbreviation ‘HFD’ in line 202, where ‘high fat diet’ appears for the first time.

Lines 944-947 I understand what the authors want to refer to. However, considering that this article will be a review article in the scientific journal with impact factor more than 4, these statements do not seem to be appropriate especially in the conclusion section. These context would better be fit into the acknowledgments.

Line 959 in Abbreviation

Please consider to add the French full name for ANSES (L’Agence nationale de sécurité sanitaire de l’alimentation, de l’environnement et du travail), as well as its English name.

Author Response

(The authors gave the same response as above.)

Round 2

Reviewer 1 Report

I appreciate the efforts made by the authors to modify the manuscript.  The manuscript is really a lot about the production methods of laboratory animal diets and I suggest this be part of the title.  Perhaps "Some comments on production methods of rodent diets used in biomedical research" 

I suggest changing …with special emphasis on "hepatology research" to "non-alcoholic fatty liver disease research” given hepatology is a broad area of research.

Line 55 – suggest replacing "hepatology" or adding after this, "in particular, non-alcoholic fatty liver disease (NAFLD)".

Line 99 – 101 – suggested sentence change - "rodent diets may require sterilization techniques when animals sensitive to normal or opportunistic microbes, such as immune compromised or germ free mice, are investigated."

Line 103 – after atmosphere-packed, in parentheses (i.e. nitrogen-purged)

Line 118 – rather than saying “Mouse diets”, this should be “grain-based diets” because this type of testing really applies only to those types of diets as their ingredients are not as refined as purified ingredients.

Line 134 – Replace “mouse diets” with “grain-based diets” and “facility“ should be “facilities”

I recommend adding section with title "3.1 Grain-based diets"

Line 137 – 139 – suggest that this sentence is modified to: Depending on the ingredients, fixed formulas can reduce variation of nutrients from batch-to batch.  However, grain-based diets may still be subject to variation due to the complex nature of the ingredients in these diets, which contain multiple nutrients and non-nutrients known to be subject to variation.

Line 152 – replace “standard diets” with “grain-based diets”

Line 183 – section should be called 3.2 Purified diets

Lines 203 –206 – suggest these 2 sentences start off the proposed section 3.1.

Lines 206 – 212 – suggest this section starts off proposed section 3.2. called "Purified diets"

Line 206 – replace purified “chows” with purified “diets”

Line 210 – replace “chows” with “diets”

Line 227 and line 228 – replace “mice” with “mouse”

Line 322 – replace “mouse diets” with “purified diets” as the ingredients described in this section are all purified ingredients

Line 330 – replace “mineral” with “minerals”

Line 334 – after “temperature” add “or cooler, depending on the composition of the diet,”

Table 2 – font size of fiber sources looks larger than the rest

Line 358 – first words “White egg” replace with “Egg white”

Line 377 – replace “mouse diets” with “purified diets”

Line 384 – after “animals”, add a comma

Line 385 – replace “produce” with “produces”

Line 392 – replace “resulted” with “can result”

Line 396- replace “in” with “and”

Line 397 – replace “the constituent” with “its constituent”

Line 397 – replace “1.597” with “15.97” and “(0.38 kcal/g)” with “(3.8 kcal/g)”

Line 397 – replace “Based” with “Depending”

Line 398 – replace “starch degradation partially resists” with “certain glucose polymers may resist”

Line 399 – replace “arrives” with “arrive”

Line 408 – replace “10% fat” with “10 kcal% fat” and “45 – 60%” with “30 – 60 kcal% fat”

Line 412 – replace “containing” with “contains a”

Line 413 – after “saturated fatty acid” add “(~30%)” and remove “triglycerides” as most fat sources contain triglycerides

Line 413 – replace “no” with “<1%”

Line 419 – replace “significant” with “significantly”

Line 420 – explain what “carbon tetrachloride” is for the reader

Line 429 – instead of “In line” maybe use a different term such as “In addition” since this was also used in the previous sentence

Line 448 – should include some data for how NAFLD is affected by menhaden oil, such as the paper by Depner et al.  J. Nutr. 142: 1495–1503, 2012.

There are 2 sections on corn oil (6.3.2 and 6.3.5) – please combine information into one section

Line 477 – replace “period” with “time”

Line 486 – replace “typical values of destruction of vitamins” with “reduced levels of vitamins”

Line 488 – replace “mouse diets” with “grain-based diets” – I believe this is more the case because fish diets typically contain grain-based ingredients

Line 522- replace “into a” with “some of which form a” – because not all soluble fibers are viscous (i.e. inulin)

Line 531 – 532 - Suggest that this sentence starting with “Similar” be revised to - "ADF is the fraction of fibers that contain virtually no fermentable ability and reduces overall digestible energy from the diet and NDF is a measure of most of the fiber in the diet (except for soluble fiber, which is not part of this fraction).

Line 568 – replace “actual” with “most recent edition, published in 1995,”

Line 586 – replace “mice” with “mouse”

Line 592 – 593 – replace “decides about” with “determines”

Line 612 – capitalize “environmental”

Line 624 – remove “these procedures” – not needed “…animal suffering require permission”

Line 639 – after “research” say “in particular, non-alcoholic fatty liver disease or NAFLD”.

Line 650 – replace “These 4 diets” replace with “These 4 diets are some examples of diets commonly used”

Line 651 – 652 – The description of NAFLD and NASH should be clarified - NAFLD is a spectrum of diseases which consists of NASH, not a progressive form of NASH.  https://gi.org/topics/fatty-liver-disease-nafld/

The reference by Machado (129), first sentence in abstract states:

Non-alcoholic steatohepatitis (NASH), the potentially progressive form of nonalcoholic fatty liver disease (NAFLD), is the pandemic liver disease of our time.  Therefore, NASH, not NAFLD, is a progressive form in the spectrum of NAFLD.

Line 657 – Rather than “NASH/NAFLD” use NAFLD/NASH, as in Machado et al, because NAFLD is the overall disease spectrum, and NASH is part of this spectrum, so NAFLD should be put first.

Line 659 – either keep NAFLD or NASH (NAFLD would cover the entire spectrum, so you can just include this)

Line 662 – Instead of just “develop” I suggest “propensity to develop”

Line 663 – instead of just “diet” I suggest “this type of diet”

Line 664 - Rather than saying hypertension, I would suggest elevated blood pressure as mice fed only high fat diet would have relatively mild increases in blood pressure.  Reference Nizar et al doi: 10.1152/ajprenal.00265.2015

Line 668 – instead of “mice background” say “genetic background” or “mouse strain”

Line 671 – before “high” include “a” and after “sugar”, I suggest “as sucrose or fructose”

Line 675 – 676 – suggest revision to "ghrelin activation requires acetylation of its 3rd residue, serine, with caprylic acid by ghrelin O-acyltransferase."

Line 692 – requires a reference Please supply a reference for this statement.  The one paper by Sumyoshi et al shows sucrose can drive glucose intolerance. https://academic.oup.com/jn/article/136/3/582/4664210

If there is a reference for fructose diet, please include this.

Line 719 – replace “add” with “be”

Line 748 – 751 – I would include this as part of the mechanistic Figure 10

Line 761 – suggest adding after reference “139” “and this trend has continued as demonstrated by a more recent survey of a larger sampling of the same journals (11).”

Line 801 – remove “was” and replace “significant” with “significantly”

Line 802 – suggest replacing “impact on” with “affecting”

Line 803 – 804 – remove “more previous”. Also, please provide the basic findings of this study as previous statements provided phenotypical changes. 

Line 807 – what concentration of daidzein?  Should be added to be consistent with previous paragraph which stated concentrations.

Line 812 – my suggestion is to include discussion of cholesterol and cholic acid diets with NAFLD diet section

Line 827 – replace “elemental” with “essential”

Line 836 – replace “and are produced” (just before “after consultation”) with “and are formulated”

Line 883 – 884 – Sentence starting with “They allow distinguishing” – I suggest saying that “They allow the researcher to distinguish one diet from another…”. 

Line 966 – suggest revision of title to “Diversity of diet ingredients may confound data interpretation”

Line 983 – add “a purified” before “HFD”, replace “normal chow” with “grain-based diet” and add “(i.e. purified diets)” after “compositionally defined diets”.

Line 991 – replace “mice” with “animal” – “laboratory animal science”

Line 991 – replace last “mice” with “mouse”

Line 992 – replace “animals” with “they”

Line 993 – replace “standard and customized” with “grain-based and purified”

Line 994 – before “are certified”, state “some are certified” (in fact, many diets are not ‘certified’)

Line 996 – Replace “Besides the actual ingredient of a chow” with “Besides modification of the different ingredients of a diet”

Line 1008 – remove “to customers”

In graphical abstract (line 1503 – 1505) – replace “standard chow” with “grain-based diet”

Replace “special diet” with “special ingredient diet”

State “Purified diet” rather than “Purified chow”

Author Response

Dear reviewer,

many thanks for reviewing the revised version of our paper. Our comments to your suggestions are given in the attached pdf-file.
